# Orthogonal LoxPsym sites allow multiplexed site-specific recombination in prokaryotic and eukaryotic hosts

Charlotte Cautereels[1,2], Jolien Smets [1,2], Jonas De Saeger [3,4], Lloyd Cool [1,2,5], Yanmei Zhu[1,2], Anna Zimmermann [1,2], Jan Steensels [1,2], Anton Gorkovskiy [1,2], Thomas B. Jacobs[3,4] & Kevin J. Verstrepen [1,2] ✉

Site-specific recombinases such as the Cre-LoxP system are routinely used for genome engineering in both prokaryotes and eukaryotes. Importantly, recombinases complement the CRISPR-Cas toolbox and provide the additional benefit of high-efficiency DNA editing without generating toxic DNA double-strand breaks, allowing multiple recombination events at the same time. However, only a handful of independent, orthogonal recombination systems are available, limiting their use in more complex applications that require multiple specific recombination events, such as metabolic engineering and genetic circuits. To address this shortcoming, we develop 63 symmetrical LoxP variants and test 1192 pairwise combinations to determine their cross-reactivity and specificity upon Cre activation. Ultimately, we establish a set of 16 orthogonal LoxPsym variants and demonstrate their use for multiplexed genome engineering in both prokaryotes (*E. coli*) and eukaryotes (*S. cerevisiae* and *Z. mays*). Together, this work yields a significant expansion of the Cre-LoxP toolbox for genome editing, metabolic engineering and other controlled recombination events, and provides insights into the Cre-LoxP recombination process.

Site-specific recombination has become a staple tool in today's molecular biology and genetic engineering in both prokaryotes and eukaryotes (see for example[1–10]). Site-specific recombination techniques mostly depend on recombinases that recognize and recombine specific DNA sequences, resulting in deletion, inversion, integration and translocation of large chunks of DNA[11]. For several genome engineering applications, site-specific recombinases have been largely replaced by CRISPR/Cas-based tools. However, recombinases still remain the prime tool in various genetic engineering strategies since they offer important advantages. For instance, recombinases do not generate DNA double-strand breaks, which are toxic to the target cell and can trigger undesirable point mutations and structural variation[12–14]. Moreover, recombinases do not rely on the presence of an efficient native homology directed repair pathway in the target organism, which is absent in various bacterial, yeast, plant and human cell types[15–19]. Therefore, recombination-based strategies are still routinely used to recuperate marker genes after transformation[20–24], introduce genomic constructs at specific loci[9,25] or enable large genomic deletions of more than 25 kb[26–28].

Apart from the use of recombinases in more traditional genetic engineering applications, novel recombinase-based engineering strategies are also being developed. For example, recombinases are at the core of the SCRaMbLE technology, developed in the Sc2.0 project to generate complex structural variation via the insertion of multiple

[1]VIB Laboratory for Systems Biology, VIB-KU Leuven Center for Microbiology, Leuven 3001, Belgium. [2]CMPG Laboratory of Genetics and Genomics, Department M2S, KU Leuven, Leuven 3001, Belgium. [3]Department of Plant Biotechnology and Bioinformatics, Ghent University, Technologiepark-Zwijnaarde 71, 9052 Ghent, Belgium. [4]VIB Center for Plant Systems Biology, Technologiepark-Zwijnaarde 71, 9052 Ghent, Belgium. [5]Laboratory of Socioecology and Social Evolution, KU Leuven, Leuven, Belgium. ✉e-mail: kevin.verstrepen@kuleuven.be

recombination sites into a synthetic *Saccharomyces cerevisiae* genome[4,29]. Additionally, recent recombinase-based engineering strategies combine prime editing with site-specific recombination to enable insertion of large fragments in the host genome without the generation of double strand breaks[30–32].

A plethora of different site-specific recombinase systems have been described and generally, all site-specific recombinases fall within one of two groups: serine recombinases (e.g., ϕC31 and Bxb1) and tyrosine recombinases (e.g., Flp, λ and Cre). Although the recombination mechanism of both groups differs, they both rely on recognition site alignments to enable DNA breakage and repair[11,33,34]. One of the most commonly used systems is the bacteriophage P1 Cre tyrosine recombinase. The Cre recombinase works efficiently and, unlike many other site-specific recombinases, without any accessory proteins[11,35–37]. Moreover, it has a high activity and is functional in a wide range of pro- and eukaryotes. Importantly, the recognition site, referred to as LoxP, is long enough to be unique even in larger genomes[11,38–41]. The LoxP site is a 34 bp sequence comprising two 13 bp inverted repeats that flank a directional 8 bp spacer[42]. Two Cre recombinase enzymes bind as a dimer to the inverted repeats of a LoxP target site. This dimer can interact with a Cre recombinase dimer bound to another LoxP site that is oriented in an antiparallel fashion to the first, resulting in a tetrameric complex that includes two active and two non-active Cre proteins (Fig. 1a). During recombination, this synaptic complex forms a covalent intermediate conformation wherein the tyrosine residues of the active Cre recombinases make a phospho-tyrosine linkage with the so-called scissile phosphates, which belong to the 1st/8th nucleotide of the spacer sequence. Secondly, an almost planar Holliday junction intermediate arises after the free 5′ hydroxyl group attacks the 3′ phosphotyrosine linkage of the other strand and initiates strand exchange. In the next step, the complex isomerizes such that the Cre proteins which were initially active are now inactive and vice versa, and recurrence of cleavage and exchange steps on the other strands finally results in a recombined DNA structure[11].

The natural Cre system is directional, with the orientation and position of the two targeted LoxP sites dictating whether Cre activity results in deletion, inversion or translocation of the DNA fragment located in-between the two LoxP sites (Fig. 1b)[11]. To enable non-directional recombination, where the type of recombination event varies stochastically and independently from the orientation of the recombination sites, a symmetrical LoxP site, referred to as the LoxPsym site, was developed (Fig. 1b). Here, the natural spacer sequence of LoxP is converted into a palindromic sequence by editing the first half of the spacer[43]. Non-directional recombination sites generate more variation in recombination outcomes, which is desirable for some applications, for example when generating genetic (and phenotypic) diversity using SCRaMbLE to screen for improved phenotypes[44–46]. Interestingly, non-directional target sites have recently also been developed for the Vika and Dre tyrosine recombinases[47].

Due to their high specificity, site-specific recombinases are highly valuable for genome engineering[1]. However, this site-specificity also implies that upon induction of recombination, all recombination sites present in a genome interact with each other in an unpredictable, stochastic fashion[1,11,33,34]. This limits the use of site-specific recombinases for genome editing, which often requires multiple, independent, specific genomic edits. Several attempts have been made to obtain orthogonal recombination systems, where recombination only occurs between specific recombination sites. Orthogonality allows simultaneous, large-scale and independent gene recombination in different regions of the genome, as well as repeated cycles of recombination that are independent of the presence of any other recombination site from a previous cycle. Such orthogonal recombination systems not only enable more sophisticated genome engineering in synthetic biology, but also have applications in other fields, such as

developmental biology[48], metabolic engineering[44], DNA assembly[49] and environmental monitoring[50].

Previous attempts at obtaining orthogonal recombination systems have relied on combining different site-specific recombinase enzymes that do not show cross-reactivity to each other's recognition sequence. For example, several natural recombinases derived from other organisms have been reported to work orthogonally from the Cre-LoxP system, including Bxb1[51], ϕC31[51], Flp[51,52], SCre and VCre[53], Vika[54] and Panto and Nigri[55]. In a different approach, directed evolution has been used to generate new Cre variants that recognize different DNA target sites[56]. These recombinases have been further developed into orthogonal split-recombinases for enhanced control over their activity[57]. However, the number of non-cross-reacting recombinases remains limited to a handful of systems. Moreover, using several recombination systems within one host organism requires heterologous expression of various enzymes, which is labor-intensive and potentially toxic to the host[58,59]. Alternative efforts have focused on altering the recombination site itself to obtain non-cross reactive (orthogonal) sites[43,47,60–64]. However, the scale of these studies was limited and typically yielded only one or two orthogonal variants, which is insufficient for many applications, for example during metabolic engineering of novel strains, when insertion of a full pathway (several genes) is often required. Orthogonal recombination sites could facilitate strain construction by enabling multiple cycles of marker regeneration or large genomic insertions using prime editing, without the generation of toxic double strand breaks or the risk of large genomic rearrangements caused by repeated usage of the same recombination site.

In this study, we aim to overcome the lack of orthogonal recombination systems by developing a large set of LoxPsym variants without cross-reactivity between each other. We characterize 63 LoxPsym sites obtained by editing the spacer sequence, and validate 1192 interactions between these variants in the model organism *S. cerevisiae*. This ultimately yielded a set of 16 fully orthogonal LoxPsym sites. We demonstrate the use of these sites for efficient multiplexing in genome engineering and show that the orthogonal LoxPsym variants are also functional in bacterial (*Escherichia coli*) and plant (*Zea mays*) models, showcasing the universality of this toolbox. By enabling site-specific recombination at multiple loci simultaneously, without cross-reactivity, our set of LoxPsym sites has the potential to become a staple tool in DNA double-strand break-free genome editing.

## Results

### The LoxPsym spacer affects recombination efficiency in yeast

To develop orthogonal Cre-LoxP recombination sites, we systematically edited the spacer sequence in the LoxP recombination site. We reasoned that the spacer was the ideal target for editing, as it participates in strand exchange during Cre recombination, and non-identical spacers have previously been shown to inhibit recombination[43,60,65]. Since the central two bases have been reported to be vital for recombination[60], we only altered the first and last three spacer nucleotides, while keeping the spacer palindromic (Fig. 1c, Supplementary Data 1, LoxPsym). We annotate the different sites as "LoxPsym-NNN", with the N referring to the sequence of the first three bases of the spacer.

A fluorescent reporter assay was set up to determine the efficiency and orthogonality of recombination of all 64 possible LoxPsym variants. In brief, a fluorescent reporter construct flanked by two LoxPsym sites was integrated into the yeast genome, and the fraction of cells that had lost fluorescence through Cre recombination was used as a proxy for the recombination efficiency (Fig. 1d, Supplementary Fig. 1a, b). It should be noted that inversion of these constructs would still display as fluorescent cells, causing us to underestimate the actual recombination efficiency. However, previous research has reported that the inversion frequency of LoxPsym is negligible compared to the

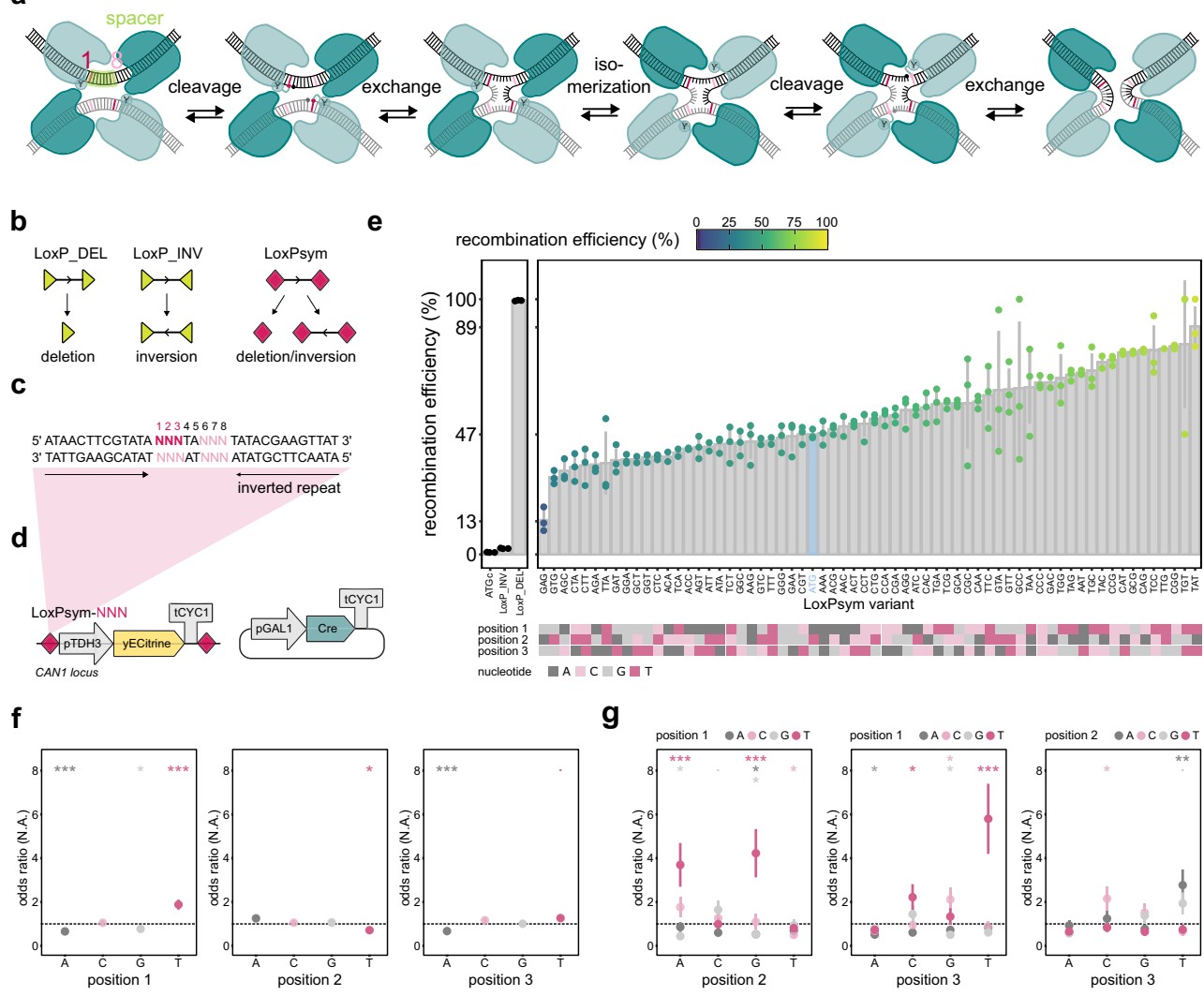

**Fig. 1 | Effect of LoxPsym spacer sequence on recombination efficiency. a** Cre-LoxP recombination. Cre monomers form a tetrameric complex comprising two active (light) and two inactive (dark) units interacting with two LoxP targets (scissile base pairs in pink). **b** The orientation of LoxP (green triangle) determines the recombination outcome (left), whereas LoxPsym (pink diamond) recombination stochastically results in deletion or inversion (right). **c** LoxPsym-NNN variants were obtained by editing spacer nucleotides 1, 2 & 3 (dark pink). Complementary nucleotides of the second strand, and complementary spacer positions 8, 7 & 6 (light pink) were modified accordingly. **d** The (genomically integrated) fluorescent reporter construct (*TDH3* promoter − *yECitrine* − *CYC1* terminator) used to determine recombination efficiency. The plasmid encoding *GAL1* promoter −*Cre* − *CYC1* terminator allowed galactose-inducible recombination. **e** Recombination efficiency of LoxPsym variants after 6 h induction. LoxPsym variants are represented by their spacer nucleotides 1, 2 & 3. Three control samples (black, left) indicate reporters flanked by LoxPsym-ATG sites with C and G at spacer positions 4 & 5 ('ATGc'),

original LoxP sites in inverted ('LoxP_INV') and same ('LoxP_DEL') orientation. Bars and error bars represent average and standard deviation of three biological replicates, respectively. Reference LoxPsym-ATG in blue. The map (bottom) graphically represents spacer nucleotides 1, 2 & 3. **f** Effects of single nucleotides on recombination efficiency for each position of the spacer, calculated by generalized linear mixed-effects model fit 3 (Supplementary Table 1), based on data of three biological repeats shown in (**e**). Odds ratios >1 (dotted line) indicate events are more likely to occur as the predictor increases, odds ratios <1 indicate the opposite. Dots and error bars represent average and standard error, respectively. Statistics by multiple pairwise-comparison by two-sided Tukey honest significant differences test on the log odds ratio scale ('***'$p < 0.001$, '**'$p < 0.01$, '*'$p < 0.05$, '.'$p < 0.1$, see Supplementary Data 1, GLME for exact *p* values). **g** Effects of interactions between nucleotides at two positions of the LoxPsym spacer. Data and statistics similar to (**f**). Source data are provided as a Source Data file.

deletion frequency in vivo[43] and in this study as well, the observed fraction of inversions was low (2.98 ± 0.628%) compared to the rest of the population (Supplementary Fig. 1c−f). To decide on the duration of the induction of recombination, we tracked the recombination efficiency over time, using a randomly selected LoxPsym variant (LoxPsym-TTA, Supplementary Fig. 1f). The results show that the fraction of cells in which the reporter is deleted through recombination gradually increases over time. To prevent saturation of these deletions and allow for comparing recombination efficiencies of different cell populations,

we decided to use an induction time of 6 h, as this showed a moderate level of recombination (55.3 ± 15.1%).

Analysis of the recombination efficiencies caused by different LoxPsym variants led to several important conclusions. Firstly, we corroborate previous findings by Hoess et al. 1986 that, for the reference LoxPsym-ATG (shown in blue), replacing the nucleotides T and A at positions 4 & 5 of the spacer by C and G (LoxPsym-ATGc) prevented recombination from occurring (Fig. 1e, left panel)[43]. Secondly, we found that the recombination efficiency differed greatly between the

## Table 1 | Analysis of deviance table of generalized linear mixed-effects model fit 3

**Analysis of Deviance Table**

| Variable[a] | Chisq | Df | *P*-value[b] | Significance[b] |
|---|---|---|---|---|
| (Intercept) | 0.0023 | 1 | 0.9615596 | |
| pos1 | 6.3514 | 3 | 0.0957096 | . |
| pos2 | 3.8416 | 3 | 0.2790817 | |
| pos3 | 6.7135 | 3 | 0.0816128 | . |
| pos1:pos2 | 48.5859 | 9 | 1.984e-07 | *** |
| pos1:pos3 | 37.5621 | 9 | 2.088e-05 | *** |
| pos2:pos3 | 29.3700 | 9 | 0.0005612 | *** |

[a]Factors pos1, pos2 and pos3 represent the nucleotides at positions 1, 2 & 3 of the spacer respectively, while interaction factors pos1:pos2 and pos1:pos3 represent the interactions between the nucleotides at these positions. The response variable is the recombination efficiency.

[b]Statistics by Type III Wald chisquare tests. Significance codes: '***'$p < 0.001$, '**'$p < 0.01$, '*'$p < 0.05$, '.'$p < 0.1$.

64 different LoxPsym variants, covering a range of 13.4% to 89.3% (Fig. 1e, right panel). Interestingly, 37 of the LoxPsym variants showed a higher recombination efficiency compared to the reference LoxPsym-ATG sequence, with the best performing site, LoxPsym-TAT, showing 1.9-fold more recombination. To confirm that the observed changes in fluorescence were indeed caused by recombination (deletion of the fluorescence marker), PCR was performed on 100 single clones of two independently induced populations, one with the highly active LoxPsym-TAT (89.3 ± 7.81%) and one with the moderate LoxPsym-CCA (53.9 ± 2.61%) (Supplementary Fig. 2). We observed a perfect correspondence between PCR fragment length and fluorescence measurements, indicating clones only lost fluorescence when a deletion was detected via PCR. These results thereby firmly establish fluorescence measurement as a fast and reliable read-out for recombination efficiency.

To better analyze the link between spacer sequence and recombination efficiency, we developed a generalized linear mixed-effects (GLME) model. The results revealed that the recombination efficiency is largely determined by complex interactions between all spacer nucleotides, most importantly the interactions between nucleotides at positions 1 & 2 and at positions 1 & 3 (Table 1, Supplementary Fig. 3, Supplementary Table 1, Supplementary Data 1, GLME). While C does not significantly affect recombination efficiency at any position, introducing G at position 1, A at position 1 or 3 and T at position 2 significantly lowered the efficiency, while T at position 1 or 3 resulted in an increased efficiency (Fig. 1f, Supplementary Fig. 3d). The enhancing effect of T at position 1 and 3 was most pronounced when it was combined with a purine (A or G) at position 2 or a pyrimidine (T or C) at position 3 (Fig. 1g, Supplementary Fig. 3e). Lastly, interactions between positions 1 & 2 showed that the recombination efficiency decreased in the presence of neighboring pyrimidines or purines at these locations. Collectively, these results indicate that the scissile base pair (position 1) and its interactions with the other nucleotides are the most important determinant of recombination efficiency. This might, at least partly, be due to the central role that the scissile base plays in several steps of the recombination process, most importantly the formation of the synaptic complex and the positioning of the DNA strands[66–68].

### LoxPsym spacer editing yields 16 fully orthogonal variants

After confirming that editing the spacer of LoxPsym yields efficient recombination, we aimed to identify combinations of fully orthogonal LoxPsym sites, i.e., different spacers that only reacted with LoxPsym sites with exactly the same spacer sequence, without showing cross-reactivity. Therefore, we systematically measured the recombination efficiency for 1192 combinations of 49 selected LoxPsym sites using a

pairwise interaction assay similar to the one described above. These included all sites with A, G and T at position 1 of the spacer, since the model revealed that these nucleotides had the strongest effect on recombination at this position (Fig. 1f). One spacer variant with a C at position 1, namely LoxPsym-CAC, was also included. The results reveal two interesting patterns (Fig. 2a). Firstly, sites that only differ at the scissile base pairs (i.e., positions 1 & 8) of the spacer sequence (e.g., LoxPsym-TAC and LoxPsym-AAC) show strong cross-reactivity, as illustrated by the diagonal lines of high recombination in the matrix of Fig. 2a. Using Sanger sequencing, it became clear that—following recombination—spacers with a mismatch at positions 1 & 8 consistently yielded a hybrid LoxPsym variant that contained traces from both parent sites and for which the nucleotides at positions 1 & 8 were no longer each other's complement (Fig. 2b). These findings agree with the hypothesis that the outer nucleotides of the spacer do not participate in strand exchange during recombination[69]. In contrast, cross-reactivity significantly decreased when there was a mismatch at position 2 and/or 3 of the spacer (Fig. 2c). In the events where cross-reactivity was observed when a mismatch at position 2 or 3 was present, sequence analysis showed that recombination sites were always restored to one of the original sequences, so that the recombined LoxPsym site was palindromic at those positions. Importantly, the nucleotides at positions 1 & 8 do play a crucial role in determining cross-reactivity between LoxPsym variants when a mismatch occurs at position 2 and/or 3 of the spacer. For instance, a A-C mismatch at position 3 between LoxPsym-TTC and LoxPsym-GTA imposed pairwise orthogonality, whereas this was not the case for LoxPsym-ATC and LoxPsym-ATA, which show a relatively high cross-reactivity of 10.27 ± 1.810% (Fig. 2a).

In addition to the high cross-reaction between variants that only differed in the 1st/8th spacer position, several other sites also showed cross-reactivity. This is in contrast to previous reports finding that sequence identity between recombining substrates is required in vitro[60], but in agreement with the findings reported by Sheren et al. 2007, who also show that in vitro and in vivo experiments do not necessarily correspond[64]. In general, for the pairs for which we observed cross-reactivity, A-T mismatches were the most abundant, whereas G-C mismatches were the least abundant, probably caused by the more complex interaction for G/C pairs (three hydrogen bonds) as compared to A/T pairs (two hydrogen bonds) (Fig. 2d). Cross-reactivity was significantly stronger when at least one of the interacting LoxPsym variants had an A at position 1 (Fig. 2a, Supplementary Fig. 4). Out of the 63 cross-reactive LoxPsym pairs that could be detected outside of the diagonal lines, 57 included at least one LoxPsym variant with an A at position 1 of the spacer, indicating that this specific spacer edit might interfere with the recombination process (Fig. 2e). Furthermore, all LoxPsym-AAN variants interacted regardless of the nucleotides at position 3, again indicating that adenine might influence the fidelity of Cre recombination (Fig. 2a). Notably, the total frequency of T involved in cross-reactions was lower than expected by chance (0.33, 0.25 and 0.25 for positions 1, 2 and 3 respectively), and this at each position of the spacer (Fig. 2e, multinomial test with $p = 2.021e{-}06$, 0.04660, 0.0004717 for positions 1, 2 and 3, respectively).

Importantly, our assay identified a set of 16 orthogonal sites, i.e., sites that differ with at least one nucleotide in position 2 and/or 3 and show no cross-reactivity (Supplementary Fig. 5a, b). This set spanned a broad range of recombination efficiencies (Supplementary Fig. 5c). While this set already provides a powerful tool for engineering (cf. further), we assessed whether this set could be further expanded by generating longer LoxPsym variants (36, 38 and 40 bp). We reasoned that extending the spacer to 10, 12 and 14 bp, respectively, might yield functional LoxPsym sequences because the prime region for recombination site recognition and interaction (the inverted repeats) remained unaltered (Supplementary Fig. 6a). We tested two sequences for each expanded site, starting from the less active LoxPsym-TTA and

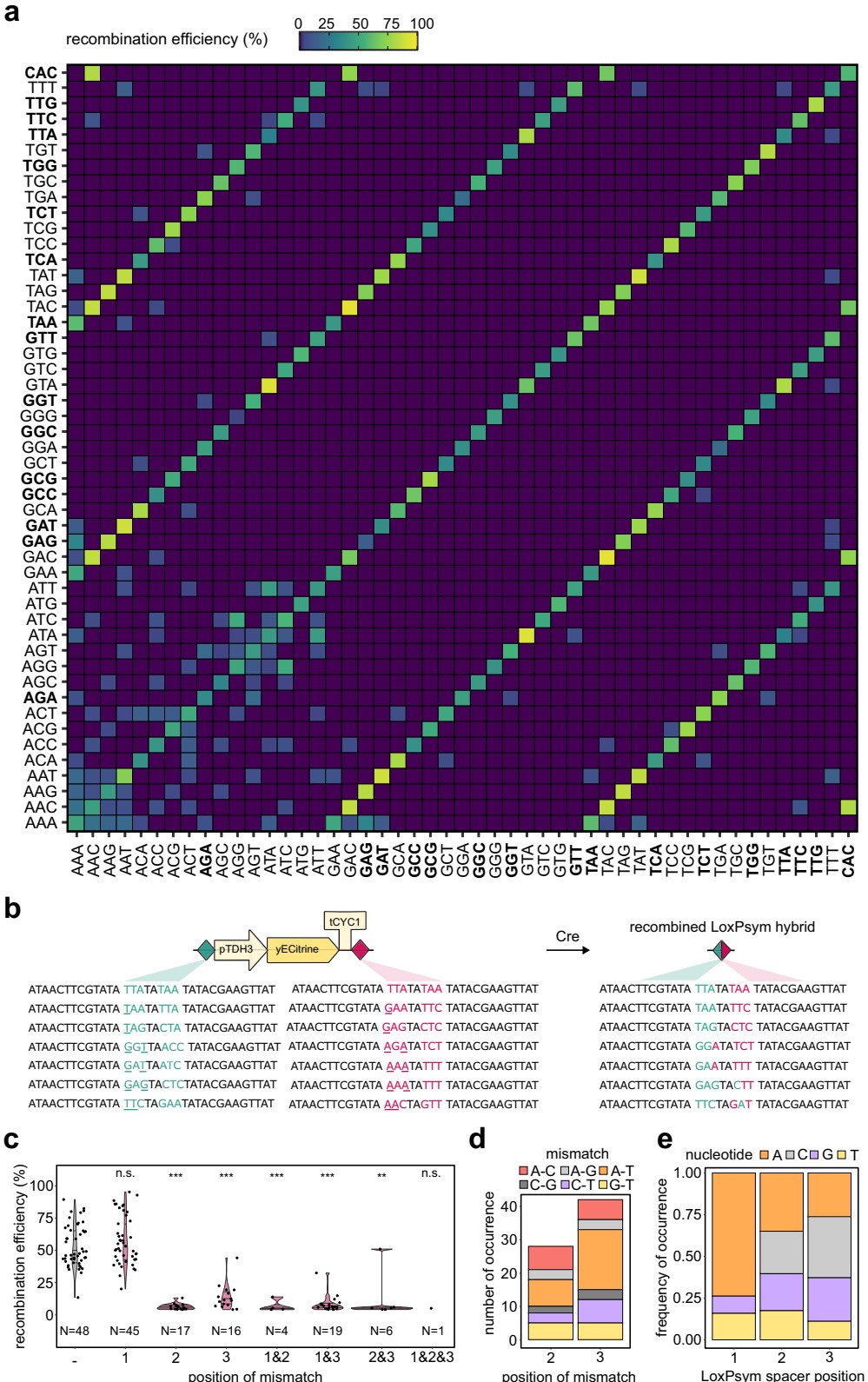

more active LoxPsym-GAC. However, only one variant (based on LoxPsym-TTA with a length of 36 bp) still showed recombination (Supplementary Fig. 6b). The inability of recombination for the other sequences did not seem to be caused by the necessity of a TA sequence at positions 4 & 5 (Supplementary Fig. 6c). Moreover, editing each position in the 10 bp spacer of the extended LoxPsym-TTA site revealed that a mismatch at every site decreased recombination, but

only a dual mismatch at position 1 & 2 completely prevented recombination from occurring (Supplementary Fig. 6d). These findings indicate that using elongated LoxPsym variants to enlarge the set of non-cross-reactive sites is not possible, most likely caused by an altered Cre-LoxP interaction that interferes with the recombination process in a sequence specific manner and does not necessarily elongate the region of strand exchange.

**Fig. 2 | Cross-reactivity between different LoxPsym variants. a** Interaction matrix representing the recombination efficiency (measured as the loss of fluorescence in the population) between different LoxPsym variants, identified by the nucleotides at positions 1, 2 & 3 of the spacer. Data represent the average of at least 2 biological repeats. Experimental set-up identical to Fig. 1d. **b** Recombination between two different LoxPsym variants resulted in hybrid LoxPsym sites (right), determined by Sanger sequencing. Underlines at the left indicate the position of the mismatch between both variants. Colors indicate the origin of the nucleotides. **c** Effect of (position of) the mismatch(es) between the LoxPsym variants (x-axis) on the recombination efficiency (y-axis). Horizontal lines in the violins represent the first quartile, median and third quartile. Statistics by comparisons to the group without mismatches (most left, gray) using two-sided Wilcoxon rank sum exact test with $p = 0.4146, 1.00e-14, 1.10e-10, 5.20e-05, 9.30e-15, 0.001160, 0.08696$ (from left to right). **d** Counts of mismatch types occurring between positions 2 or 3 of cross-reacting LoxPsym variants. **e** Frequency of nucleotides present at each position of cross-reactive LoxPsym variants. Statistics by multinomial test with $p = 2.021e-06, 0.04660, 0.0004717$ for positions 1, 2 and 3, respectively. Source data for this figure are provided as a Source Data file.

## Orthogonal LoxPsym variants for multiplex genome engineering

Next, we investigated the capability of simultaneously using all 16 orthogonal LoxPsym variants that were identified in the previous assays. Specifically, we wanted to assure that even when all 16 sites were present in the same genome, orthogonality was maintained, which is essential when these recombination sites would be used, for example, to facilitate complex metabolic engineering efforts where typically several genomic loci are altered for gene insertion or deletion, either simultaneously or consecutively. Our test relied on 18 constructs, including 16 test constructs that each assessed the functionality of one LoxPsym variant in the presence of all other variants, and 2 controls (Fig. 3a). To select for recombination in all cases, all constructs used the deletion of *URA3* (resulting in tolerance towards 5-fluoroorotic acid; FOA) by surrounding the marker with two LoxPsym-TCA sites and plating on SC+FOA. Additionally, to determine efficiency of each specific LoxPsym site in the presence of all other sites, each construct encoded an *ADE2* marker. The deletion of this marker results in a red pigmented yeast colony and allows the detection of a second recombination event. The two control constructs (differing in the location of the *ADE2* marker) exhibited only one copy of each LoxPsym site (in exception of LoxPsym-TCA, which allows for selection of recombination positive clones) and deletion of *ADE2* was not expected. In contrast, each test construct included one extra copy of one specific LoxPsym variant upstream of the *ADE2* marker and deletion of *ADE2* was expected and used as a read-out of the recombination efficiency of that site (Fig. 3b, c, Supplementary Data 1, counts). Interestingly, LoxPsym-CAC showed a very low recombination rate ($1.8320 \pm 1.410\%$) that did not significantly differ from the control constructs (two-sided Dunnettx's multiple comparisons of means with $p = 0.3487$ and $p = 0.05595$ for comparison to control 1 and 2, respectively), indicating that recombination of this LoxPsym variant was strongly reduced by the presence of the 15 other sites. This severe decrease in activity may not be desirable for multiplexed LoxPsym applications, in which case LoxPsym-CAC could theoretically be substituted by either LoxPsym-TAC or LoxPsym-GAC. The other LoxPsym variants showed higher activities, although the recombination efficiencies were consistently lower and did not correlate well with those calculated from the pairwise interaction assay of Fig. 1e (Supplementary Fig. 7a, b). This could at least in part be caused by differences in the experimental setup. In particular, our results suggest that the genomic context of the sites plays a major role because the two sites showing the highest recombination efficiency (LoxPsym-TTA and -TCA) were located at both edges of the LoxPsym array. Moreover, a negative correlation ($R^2 = 0.38$, Pearson correlation test with $p = 0.01037$) can be observed between the recombination efficiency and the distance to the edge of the LoxPsym array, indicating that the efficiency drops because more LoxPsym variants hinder the recombination site of interest to be bound or find its correct interaction partner (Fig. 3d). We hypothesize that this may be due to a combination of the negative correlation between the recombination efficiency and the distance between interacting recombination sites[70,71], the reduced ratio of Cre enzymes and its target site and the formation of nonproductive synapses between incompatible recombination sites,

which could shield the recombination sites from recombining with the compatible interaction partner[60,72].

In addition, the design of the constructs allowed calculating the frequency of cross-reactivity by PCR, since recombination between different LoxPsym sites would result in reporters with a different length (Fig. 3e). To assure that the recombination events corresponded with the expected patterns, we sequenced three PCR fragments of the correct size (i.e., the size expected when both the *URA3* control marker and the *ADE2* marker were deleted by recombination between identical LoxPsym variants) per test construct and observed the expected result in all cases (green triangles in Fig. 3e). The measured PCR fragment length deviated from the expected size (red-ish triangles on Fig. 3e) in 54 out of the 576 randomly selected red colonies (36 per test construct), indicating some level of cross-reactivity. To investigate if we could detect patterns within these undesirable cross-reactions, all 54 fragments were sequenced (Fig. 3e, Supplementary Data 1, cross yeast). Most cross-reactions only occurred a few times, with the exception of the interaction between LoxPsym-CAC & -TTC (24 instances) and LoxPsym-TCT & -TCA (9 instances). Moreover, 21 of the cross-reactions occurred with LoxPsym-TCA, the outermost LoxPsym variant which was used for positive selection of recombination, again suggesting that LoxPsym activity is skewed by the experimental set-up and that the degree of LoxPsym insulation also affects the likelihood of specific cross-reactions to occur. Therefore, we hypothesize that other cross-reactions may have been detected if another layout of the LoxPsym array would have been used. Most importantly, when normalizing the number of observed illegitimate recombination events to the number of potential occurrences, we observe that the level of illegitimate recombination is negligible for each LoxPsym variant (Fig. 3f). Specifically, the overall frequency of illegitimate recombination is 0.0839%, indicating that this set of LoxPsym variants can be considered fully orthogonal (see Materials and Methods for calculation).

## Orthogonal LoxPsym variants in bacteria and plants

After identifying and multiplexing the set of 16 orthogonal LoxPsym variants in yeast, we next tested if these recombination sites were also functional and orthogonal in other species, specifically *Escherichia coli* and *Zea mays*. For assessment of the functionality and cross-reactivity of the LoxPsym variants in *E. coli*, we set up a plasmid-based assay testing pairwise combinations between 16 different donor and acceptor plasmids, each carrying one LoxPsym variant (Fig. 4a, b). After inducing recombination, cross-reactivity between LoxPsym variants was detected via PCR amplification of the junction that spanned the recombined recombination site. All tested LoxPsym variants showed recombination activity in bacterial cells, although no correlation was observed with the activity of the respective sites in yeast and plants (Fig. 4c, Supplementary Fig. 8, Supplementary Data 1, densitometry). In contrast to the data obtained for *S. cerevisiae*, we did observe cross-reactivity in a few cases. Sequencing of these recombined scars revealed up to three mutations in LoxPsym sites resulting from recombination between cross-reactive partners (Supplementary Data 1, cross bacteria). Actually, it has previously been observed that results indicating orthogonal recombination are not always

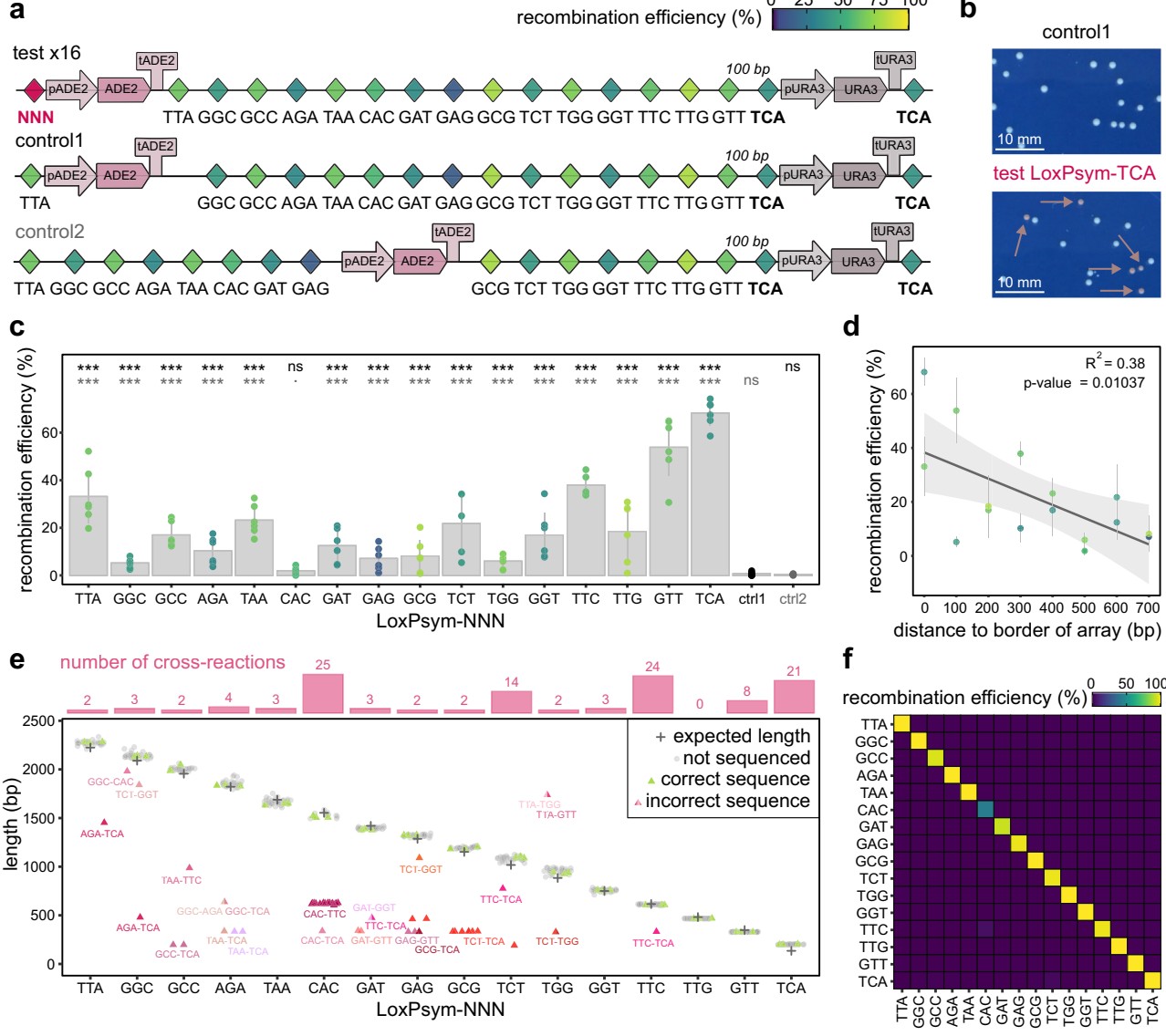

**Fig. 3 | Limited cross-reactivity between 16 LoxPsym variants simultaneously. a** Each construct (16 test and 2 control) included all 16 LoxPsym variants and *ADE2* and *URA3* expression cassettes. The *URA3* cassette was flanked by LoxPsym-TCA (selection for recombination). Controls tested 2 locations of the *ADE2* cassette and should not result in *ADE2* deletion. Test constructs differed in the LoxPsym-NNN variant upstream of *ADE2* (pink) and verified cross-reactivity between all sites and recombination efficiency between identical LoxPsym-NNN. LoxPsym variants in the array were separated by 100 bp[70]. Constructs were inserted at the *CAN1* locus of *BY4741 ΔADE2* carrying plasmid pSH47-His-Cre or pSH47-His-Vec (negative control). **b** After 6 h induction, cells were plated on SC+FOA plates (*URA3* deletion with LoxPsym-TCA). Red clones (deletion of *ADE2*) were selected for PCR and sequencing. **c** Percentage of the population with *ADE2* deletion (red phenotype). Dots represent plate counts of six biological replicates, bars and error bars indicate average and standard deviation, respectively. Color indicates pairwise recombination efficiency (Fig. 1e). Control strains showed a negligible frequency of *ADE2*

deletions (0.61 ± 0.70% (control 1) and 0.37 ± 0.22% (control 2)). Statistics by analysis of variance and two-sided Dunnettx's multiple comparisons of means ('***'$p < 0.001$, '.'$p < 0.1$, 'ns' $p > 0.1$, Supplementary Data 1, counts) to control 1 (black) and control 2 (gray). No colonies were observed for strains carrying pSH47-His-Vec (Supplementary Data 1, counts). **d** Pearson correlation test between average recombination efficiency (error bars indicate standard deviation) and degree of LoxPsym insulation (distance to array border). Colors similar to (**a**). **e** Measured (dots/triangles) and expected (crosses) length of recombined constructs of 36 randomly selected red clones from each strain. Three random samples of correct size and all fragments of unexpected size (illegitimate recombination) were analyzed (Sanger sequencing, triangles). Color indicates sequencing result (green if correct, red-scaled if incorrect, in which case interacting LoxPsym-pairs are indicated). Bars indicate the number of illegitimate recombination events per LoxPsym variant. **f** Recombination efficiencies of red clones normalized by the number of possible observations. Source data are provided as a Source Data file.

transferrable between pro- and eukaryotes, which may be caused by a slightly altered protein structure or activity of the recombinase in different host organisms, or differences in native cellular processes, such as the involved DNA mismatch repair pathway to restore mismatches that appear after recombination[61,73]. Additionally, the difference in experimental set-up can also be an explanation for the observed differences: recombination (deletion) was irreversible for the

experiments in yeast, whereas recombination of the two plasmids was reversible in bacteria. Importantly, the cross-reactivity was much lower than the recombination activity observed between identical LoxPsym sites (Supplementary Fig. 8g).

For the characterization of cross-reactivity in higher eukaryotes, we used *Zea mays*, one of the most important cereal crops with a widespread use in food and feed, as well as in industrial applications[74].

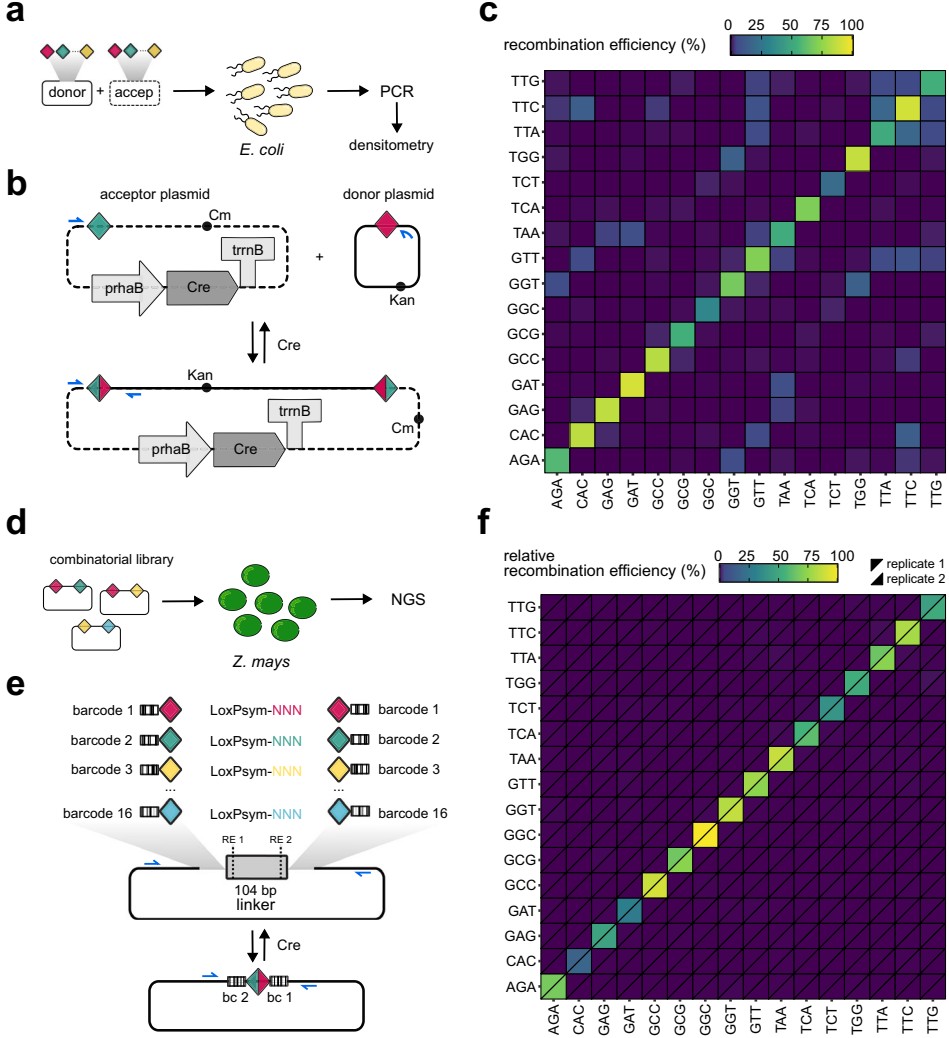

**Fig. 4 | Recombination and cross-reactivity of LoxPsym variants in *E. coli* and *Z. mays*. a** Experimental design for determining LoxPsym variant cross-reactivity in *E. coli*. A donor (full line) and acceptor (dashed line) plasmid were co-transformed, each carrying one LoxPsym variant (differently colored diamonds) and in vivo recombination was verified using PCR. **b** Acceptor plasmid encodes the *Cre* gene, controlled by rhamnose inducible *rhaB* promoter and *rrnB* terminator. Induction of *Cre* expression (4 h) results in recombination only when LoxPsym variants are cross-reactive, in which case an amplicon will be generated by PCR (blue arrows). Note that the recombination reaction does not have a final state because recombined plasmids can recombine back to separate plasmids. **c** Recombination efficiencies between LoxPsym-NNN variants in *E. coli*, calculated from densitometric analysis of the junction PCR. Data represent band intensities of PCRs performed in technical duplicate, using a mixture of templates derived from three biological replicates. **d** Experimental design for determining LoxPsym variant cross-reactivity in *Z. mays*. The combinatorial library that included all 256 pairwise combinations of LoxPsym variants was transfected to *Z. mays* protoplasts together with a plasmid

for constitutive *Cre* expression. Occurrence of recombination was verified using NGS. **e** Each plasmid of the combinatorial library encoded two LoxPsym variants (differently colored diamonds), separated by a 104 bp linker (gray) that contains recognition sites (RE1 and RE2) for restriction enzymes NcoI-HF and PvuI-HF (dashed lines). Barcodes were incorporated up- and downstream of LoxPsym variants, with each barcode uniquely linked to one LoxPsym variant. The library contained all 16 × 16 (=256) combinations between LoxPsym variants. Blue arrows indicate primer annealing sites for the PCR that was carried out after recombination induction. PCR amplicons were analyzed by NGS. **f** Recombination efficiencies between LoxPsym-NNN variants in *Z. mays*, calculated from the abundance of sequenced reads. Note that all efficiencies were normalized to the most active recombination site, LoxPsym-GGC, for which the efficiency was arbitrarily set to 100%. Data represent the average of three technical repeats for two biological replicates, shown separately by diagonally split cells. Source data are provided as a Source Data file.

We performed a plasmid-based assay using maize mesophyll protoplasts. Briefly, we constructed a library of plasmids which contained two LoxPsym sites separated from each other by a short linker that incorporated two restriction sites to digest this plasmid later in the workflow (Fig. 4d, e). Each LoxPsym variant was accompanied by a unique barcode allowing the identification of which LoxPsym pair was involved in the recombination process by sequencing barcodes surrounding the recombined LoxPsym site (Supplementary Table 2). A combinatorial cloning scheme was used, and all 256 combinations of the 16 LoxPsym variants were present in the final plasmid pool, as confirmed by NGS sequencing (Supplementary Fig. 9a, Supplementary

Data 1, NGS). The plasmid pool and a plasmid constitutively expressing the Cre recombinase or an empty backbone were co-transfected into maize mesophyll protoplasts and the region spanning the LoxPsym site(s) was amplified by PCR 48 h after transfection. Recombination was only detected in the presence of the Cre recombinase and this reaction was send for NGS sequencing to assess the efficiency of recombination of each LoxPsym pair by identifying barcode frequencies in the pool (normalizing to the abundance in the starting pool) (Fig. 4f, Supplementary Data 1, NGS Supplementary Fig. 9b, c). The results confirmed activity of recombination in the higher eukaryote *Z. mays*, and showed the absence of cross-reactivity between

selected LoxPsym variants. Moreover, a broad range of recombination efficiencies linked to different LoxPsym sites was detected, albeit with a weak correlation to the efficiencies that we observed in yeast ($R^2$ = 0.02, Pearson correlation test with $p$ = 0.6056) (Supplementary Fig. 9d).

## Discussion

In this study, we present the development and characterization of 63 LoxPsym sites in the yeast *Saccharomyces cerevisiae*. We assessed cross-reactivity between a selection of these variants by a pairwise interaction assay and identified a set of 16 orthogonal LoxPsym variants that can be used simultaneously with no or only minimal cross-reaction. Using a systematic approach for LoxPsym modification, our study reveals the following insights to Cre-LoxPsym recombination: simply modifying the recombination site spacer sequence is insufficient to impose orthogonal functioning, refuting the general assumption in the field; the scissile base pairs play an unexpected, yet crucial role in the orthogonality decision; neighboring recombination sites affect each other's efficiencies and design rules to tweak recombination site activity. We demonstrate that the sites can also be used in other species, including *E. coli* and *Z. mays*. Together, these findings dramatically expand the potential of using Cre-LoxPsym as a gene editing technology, especially for cases where recurrent and/or multiplexed recombination is desirable, for example during strain construction in metabolic engineering efforts.

We anticipate that the presented set of highly efficient and fully orthogonal recombination sites will mainly find its application in non-invasive genome engineering of living systems. On one hand, they provide an alternative to CRISPR-based systems that are subject to a complex IP regulation, which defers many commercial users from exploitation. On the other hand, their site-specificity allows for multiple, precise and efficient genomic edits without the need to rely on a cell's innate homologous recombination machinery or toxic double-strand breaks. Especially for applications of complex metabolic engineering that require insertion and deletion of multiple genes, one could use the set of LoxPsym sites to recycle selection markers. Moreover, strains or cell lines with several LoxPsym landing pads (i.e., stretches of DNA containing one or multiple different recombination sites) could be constructed, which can serve as a starting point for multiplexed engineering, thereby significantly speeding up the process of strain or cell line construction[75]. In fact, several research groups have recently exploited the utilization of recombinases for both knock-out and knock-in of large DNA constructs to escape the exposure to toxic double-strand breaks caused by homology-directed repair approaches, such as conventional CRISPR-Cas-based methods[1,30,31,76]. Repeated utilization of the same recombination site in such approaches can cause undesired structural rearrangements at different genomic loci, a problem which can be circumvented perfectly by the presented LoxPsym toolbox. To further enhance integration efficiencies at target sites, one could combine the orthogonal LoxPsym spacer sequences of the presented study with previously developed recombination site arms that have been reported to boost integration[77]. A potential downside of the Cre-LoxP system is the introduction of recombination scar sites. However, the exact location of these sites is predictable and in most instances, these scars are not problematic and can even be a benefit. For example, Yarnall et al. (2023) have turned used scar sites into a linker sequence to fluorescently tag target proteins[31].

In addition to their use for more sophisticated genome engineering in synthetic biology, our set of orthogonal recombination sites can also find applications in other fields. For instance, the Cre recombinase has been repeatedly used in the design of genetic circuits[5,78,79] and biosensors[80], and the implementation of our orthogonal recombination sites could facilitate more complex circuit designs. Also, the limited set of orthogonal recombination systems is commonly used for gene expression activation/inactivation in systems biology, for example to study aging processes[81], oncogenic mutations[2,82,83], antibiotic resistance[84], cellular heterogeneity[80], cell lineage tracing[85], and our set of orthogonal recombination sites can drastically expand the number of genes that could be controlled simultaneously. Moreover, orthogonal recombination sites can also serve as a manner to induce structural variation of defined regions at several genomic loci in parallel, to study the impact of genomic context on cellular function with enhanced control over the variant outcome. Finally, new synthetic biology tools could also implement our set of orthogonal sites to shuffle genetic elements (e.g., synthetic promoters) to rapidly generate libraries with diverse phenotypes (e.g., expression level diversification[86]).

Our tests of different LoxPsym sequences revealed that editing the spacer sequences does not prevent recombination, despite the Cre protomers making contact with the LoxPsym spacer sequences[87]. Importantly, we observe that the sequence of the spacer has a major influence on the recombination frequency, with recombination events observed in a range of 13% to 89% of the cell population. This could have several reasons, such as the altered interaction strength between the enzyme and the target sequence, the efficiency of synapsis, the ease of strand partition or the change in free energy needed for DNA bending of the spacer region during the recombination process, which stretches up to 100 degrees during the formation of the synaptic complex[66,87–89]. Further statistical analysis elucidated the relationship between the sequence and the activity of LoxPsym variants. Most importantly, we find that the scissile base pairs (positions 1 & 8 of the LoxPsym spacer) are the prime determinant of recombination efficiency. Although the scissile base pairs do not participate in strand exchange[69], they play a crucial role in several steps of the recombination process. Specifically, they directly interact with the recombinase via a base-specific linkage[87], influence the efficiency of synapsis[66], regulate the positioning of the DNA bend needed during the formation of the synaptic complex[68,90], and determine the order of strand exchange[60], although not for symmetrical recombination substrates[91].

Our analysis shows that a thymine at position 1 of the LoxPsym spacer greatly enhances the efficiency of recombination, whereas the opposite is true for adenine or guanine. This agrees with a previous observation in a limited set of non-symmetrical LoxP substrates[92] and might indicate that the presence of pyrimidine bases in the tightly compressed minor groove adjacent to the non-active scissile base of the Holliday junction[68] impedes recombination. Other studies focusing on LoxP editing did not reveal statistically significant relationships between the LoxP spacer and the recombination efficiency[60,61,63,64], suggesting that our systematic approach and analysis provides an important step towards a better understanding of the recombination process.

Interestingly, we identify LoxPsym variants that either show weaker or stronger recombination efficiencies than the canonical LoxPsym-ATG sequence[43]. Weaker recombination sites could help applications requiring moderate recombination activity. For example, high activity of recombination has been reported to increase cell lethality when SCRaMbLEing the Sc2.0 chromosomes, presumably due to the frequent deletion of essential genes[44,93]. Similarly, sparse labeling in neuroscience applications also requires recombination sites with lower recombination efficiencies to enable labeling of only a small fraction of cells in an overall population[7]. Conversely, recombination sites with enhanced activity are needed for the development of specific efficient gene editing tools. For example, highly active site-specific recombination could boost the efficiency of tools for optogenetics[36,76,94,95], cell lineage tracing[96], gene regulation[82], genetic circuit design[4,97] and gene expression cascades[3]. Moreover, the 16 orthogonal recombination sites that we identified in this study might help to further expand these existing technologies, as they can be used simultaneously without cross-reactivity.

Together, our study yielded symmetrical target sites for the Cre recombinase with high versatility and a wide range of activities, including enhanced activity to the state-of-the-art LoxPsym site. The establishment of a relationship between the LoxPsym sequence and recombination efficiency both deepens our insights into the recombination process and expands the synthetic biology toolbox. Sixteen of the identified LoxPsym variants can be used simultaneously without cross-reactions, enabling rapid and multiplexed genome engineering. Site-specific recombinases remain a pivotal tool for genomic manipulations in a wide range of organisms, either in cases where CRISPR/Cas mediated editing is hampered due to low efficiencies of homologous recombination of the host organisms or in conjunction with CRISPR/Cas based tools, for example in combination with prime editing to enable repeated insertion of large DNA constructs at defined loci. Our orthogonal LoxPsym sites have the capability to become a staple tool for the development of new synthetic biology tools or high-throughput engineering efforts.

## Methods

### General methods
DNA amplification was done by PCR using SapphireAmp Fast PCR mix (Takara Bio), Phusion (NEB) or GXL Primestar (Takara Bio) DNA polymerase. DNA oligonucleotides were obtained from Integrated DNA Technologies (IDT). A list of all plasmids and oligonucleotides can be found in Supplementary Data 1, plasmids and oligo's. Synthesis of longer DNA constructs was ordered from Qinglan Biotech, BGI (Supplementary Data 1, constructs). The pV1382 backbone (P0 in Supplementary Data 1, plasmids, Addgene Plasmid #111436) was used to express sgRNA, which was ligated into the BsmBI (NEB) digested and dephosphorylated (CIP, NEB) backbone after annealing and phosphorylation (T4 polynucleotide kinase, NEB) of the oligonucleotides[98]. Plasmids reported in this study were constructed using Gibson Assembly (NEBuilder HiFi DNA Assembly Master Mix) for plasmids used in *E. coli* and *S. cerevisiae*, and using Golden Gate cloning (GreenGate cloning standard reported by Lampropoulos et al. 2013) for plasmids used in *Z. mays*[99]. Purification of plasmids needed for experiments in yeast and bacteria were performed using the QIAprep Spin Miniprep Kit (Qiagen). Purification of plasmids needed for experiments in plant cells were done using the ZymoPURE II Plasmid Midiprep Kit (Zymo Research). Sanger sequencing was performed by Eurofins Genomics.

### Strains and growth conditions
A list of all plasmids, oligonucleotides and synthesized DNA constructs used in this study for strain construction can be found in Supplementary Data 1, plasmids, oligo's and constructs. *E. coli* strains were constructed from the lab strain DH5α (NEB) and cells were grown in Luria Bertani (LB) medium (10 g/L peptone, 10 g/L NaCl, 5 g/L yeast extract) at 37 °C, shaking at $18 \times g$. Antibiotics (chloramphenicol, carbinicilin and kanamycin) were added at 50 µg/mL. Inducer L-Rhamnose was added at 2%. *S. cerevisiae* strains were constructed from the lab strain *BY4741*, which is an *S288C*-derivative laboratory strain with genotype MATa his3Δ1 leu2Δ0 met15Δ0 ura3Δ0. Cells were grown in Synthetic Complete (SC) medium (0.79 g/L SCM, 6.7 g/L YNB) or SC-Histidine medium. Carbon sources (glucose, raffinose and galactose) were added at 2%. *Z. Mays* protoplasts (cv. B104) were isolated and suspended in W5 solution, see below[100].

### *S. cerevisiae* transformation protocol
1 mL ON culture in 2xYPD (20 g/L yeast extract, 40 g/L peptone, 4 g/L glucose) was inoculated into 50 mL 2xYPD for 3 h in flasks, an inoculation volume of 1 mL in 96 well plates was used for high-throughput transformations. Cells were centrifuged (3 min, $3019 \times g$) and consecutively washed with 10 mL and 1 mL 0.1 M lithium acetate (LiOAc). Cells were resuspended in 100 µL 0.1 M LiOAc. PCR amplified donor

DNA (50 µL) and/or plasmid DNA (200 ng) were added. CRISPR/Cas9 was used for genomic DNA integration using pV1382 with inserted gRNA of interest (gRNA sequences can be found in Supplementary Data 1, oligo's). A mixture containing 620 µL 50% PEG 3350, 4 µL salmon sperm DNA and 90 µL 1 M LiOAc was added and mixed by vortexing. Cells were incubated for 30 min at 30 °C, $18 \times g$. 100 µL DMSO was added prior to a 15 min heat shock at 42 °C. Cells were harvested by centrifugation (3 min, $3019 \times g$) and washed with 5 mM $CaCl_2$. Cells were incubated for a 3 h recovery period at 30 °C, $18 \times g$ and finally plated on selective medium. Colony PCR (SapphireAmp Fast PCR Master Mix, TaKaRa) using a template prepared by boiling the clone in 50 µL NaOH (0.02 M) (99 °C, 10 min) to amplify the junction of desired insertion was used to identify positive transformants.

### Fluorescence assay and recombination in *S. cerevisiae*
Strains were derived from *BY4741* with constitutive expression of fluorescent reporter mCherry at the *YRO2* locus[101]. The mCherry reporter was used as a control to remove non-fluorescent cells. To test the LoxPsym variants, strains carried an overexpressed *yECitrine* reporter gene, which was regulated by the *TDH3* promoter and *CYC1* terminator, flanked by two LoxPsym variants (inserted via LoxPsym-tailed primers) and genomically integrated at the *CAN1* locus (using sgRNA1 and sgRNA2, OF/R 1 and OF/R 2, Supplementary Data 1, oligo's). Single colonies were inoculated in 100 µL SC-His 2% glucose for ON growth. Cells were washed and diluted in SC-His 2% raffinose to a final $OD_{600 \, nm}$ 0.05 and grown ON. Cells with the control backbone (without *Cre*) and the plasmid with the *pGAL1-Cre* expression cassette (P1 and P2, Supplementary Data 1, plasmids) were washed and diluted in SC-His 2% raffinose 2% galactose for induction of *Cre* expression. Cells were induced for 6 h, unless indicated otherwise. Cells were washed and diluted to SC 2% glucose for ON recovery (dilution 1/20), after which cells were plated on YPD and/or used for flow cytometry analysis.

### Fluorescence analysis
Flow cytometry was performed using the Attune NxT Flow Cytometer and Auto Sampler. Cultured yeast cells were diluted in focusing fluid and measured with a flow rate of 200 µL/min. Cytometry data was gated based on the FSC-H to FSC-A map to select for single cells (Supplementary Fig. 1b). For determination of the recombination efficiency, an additional gating was performed using the control fluorescent reporter mCherry (mCherry + cells were selected for further analysis). yECitrine and mCherry were measured using channels BL1-A (excitation at 488 nm and emission at 574 nm with 20 nm bandwidth) and YL2-A (excitation at 561 nm and emission at 610 nm with 20 nm bandwidth), respectively. Analysis and gating steps were done using the FlowJo software version 10.6.2 with (non-) fluorescent control strains as a reference. Recombination efficiencies lower than the reference strain constitutively expressing yECitrine without the presence of recombination sites (4.5%) were represented by the darkest color in the figures to remove noise from the data. To determine yECitrine fluorescence of single clones, single colonies were inoculated in SC 2% glucose and fluorescence was measured using plate reader (TECAN Infinite 200 Pro), using excitation at 498 nm with bandwidth 9 nm and emission at 535 nm with bandwidth 20 nm. Data was obtained after normalization by the absorbance at 600 nm. Division into fluorescent/non-fluorescent groups was done by comparison with values obtained for control strains.

### Generalized linear mixed-effects (GLME) model development
A GLME model was developed in R to describe the relationship between the recombination efficiency (response variable) and the sequence of the LoxPsym spacer (R package lme4[102], emmeans[103] and effects[104]). Fixed effects incorporated in the model were the nucleotides present at positions 1, 2 and 3 of the LoxPsym spacer and the

interactions between all of these positions, whereas the random effects used were the replicates. Replicates were labeled numerically (1 to 192, the number of rows in the data frame) to account for the effect of overdispersion. Cell counts were used as weights. Different fits were compared using the computed Akaike's information criteria (AIC) and the best fit was selected for further improvement by eliminating included factors. Finally, fit 3 proved the best model to describe the data (Supplementary Table 1, Supplementary Fig. 3c). Significant differences between the nucleotides at each position were determined using a post-hoc analysis with emmeans.

### Multiplexed-LoxPsym assay and recombination in *S. cerevisiae*

Strains were derived from *BY4741* with a deletion of *pADE2-ADE2-tADE2*, constructed using sgRNA3 (OF/R3, Supplementary Data 1, oligo's). Test and control constructs (Fig. 3a, Supplementary Data 1, constructs) were inserted at the *CAN1* locus (using sgRNA1 and sgRNA2, Supplementary Data 1, oligo's) and either P1 (Cre) or P2 (control). Induction of recombination similar to methods described above. After ON recovery in SC 2% glucose, cells were plated on SC and SC+FOA and incubated for 48 h at 30 °C, after which colonies were counted for each plate. Red colonies were selected for PCR amplification of the recombined constructs (OF/R43, Supplementary Data 1, oligo's). Length of the amplicons was determined using capillary electrophoresis (QIAxcel Advanced instrument, QIAxcel DNA Screening Cartridge, QX Size Marker 250 bp–4 kb v2.0, QIAxcel ScreenGel version 1.1.0) to visualize small differences in band length. For each test construct, three randomly selected fragments of correct length were sent for Sanger sequencing (analysis using SnapGene version 5.2.4). All fragments that deviated from the expected length were also send for Sanger sequencing to determine the occurrence of each illegitimate cross-reaction. Recombination frequencies of all LoxPsym-combinations of the selected red clones were calculated by normalization to the total number of events that could be detected (36 for recombination between identical LoxPsym sites and 576 (the total number of analyzed samples) for cross-reactions between non-identical LoxPsym sites). The overall chance of illegitimate cross-reactivity was determined by the division between the observed number of illegitimate reactions (58) and the total number of cross-reactions that could occur (120/sample*576 samples).

### E. coli transformation protocol

For heat shock transformation, chemically competent *E. coli* cells (home-made) were thawed on ice for 30 min. Plasmid DNA (50–100 ng) or 2 µL of the Gibson/Golden Gate reaction was mixed with 25 µL of competent cells in an ice-cold 1.5 mL Eppendorf tube. After 30 min incubation on ice, the reaction was heat shocked for 30 s at 42 °C and chilled on ice for 5 min. A volume of 300 µL LB medium was added, and the tube was incubated at 37 °C for 60 min in a shaking incubator. Finally, 100 µL of cells was plated on pre-warmed (37 °C) LB medium containing the appropriate antibiotics and incubated at 37 °C for ON growth. For electroporation, we used commercial NEB 10β cells (NEB) with a transformation efficiency of $2 \times 10^{10}$ cfu/µg. 2 µL of the assembly reaction was mixed with 50 µL of competent cells and placed inside a chilled electroporation cuvette (0.2 cm gap, BioRad). The electroporation was carried out in a GenePulser (BioRad) according to the manufacturer's conditions and 900 µL of SOC medium was added immediately to the cells afterwards. Cells were incubated at 37 °C for 60 min in a shaking incubator. Finally, 100 µL of cells were plated per pre-warmed (37 °C) LB plate containing the appropriate antibiotics.

### Recombination assay in E. coli

Bacterial strains were derived from DH5α after co-transformation of acceptor (P19-P34) and donor (P35-P50) plasmids using double selective medium LB + kanamycin (Kan) + chloramphenicol (Cm). Single colonies were inoculated in 100 µL LB + Kan + Cm for ON growth. Cells

were washed and diluted (1/20) in LB 2% rhamnose + Kan + Cm for 4 h growth to induce *Cre* expression (under control of the *rhaB* promoter) from acceptor plasmids (experimental conditions based on previous reports[10,105]). After induction, cells were washed and grown ON in LB + Kan + Cm. Recovered cells were harvested by centrifugation (4109 x *g*, 5 min) and suspended in dH$_2$O. Cells were boiled for 10 min at 99 °C and the remaining mixture was used as a template for PCR to amplify the junction of recombined donor and acceptor plasmids. We reasoned that amplifying one of the two recombined junction possibilities (donor plasmid could insert in two directions in the acceptor plasmid) was sufficient, as recombination between symmetrical sites should not favor one of both options and the combination of two independent plasmids avoided the accumulation of one recombination outcome. Amplicons were subjected to densitometric analysis using the Image J software to extract peak areas (from a plot of the lanes). Peak areas of the junction were normalized by division with areas extracted from the most abundant control amplicon (derived from PCR performed on separate donor and acceptor plasmids).

### Combinatorial LoxPsym library construction for Z. mays assay

For construction of the LoxPsym combinatorial library (P51-P216 in Supplementary Data 1, plasmids), we applied Golden Gate cloning using the GreenGate cloning standard[99] to assemble 5 entry clones. The cloning scheme is depicted in Supplementary Table 3. Entries A and E were constructed by ligating annealed oligonucleotides OF/R47 and OF/R48, respectively into BsaI-digested entry vectors pGGA000 (Addgene #48856) and pGGE000 (Addgene #48860). The entries for the 16 barcode-LoxPsym combinations in position B and D were made in the same manner using oligonucleotides OF/R50-OF/R65 and OF/R66-OF/R81, respectively. The linker at position C was PCR amplified from the pUC19 plasmid (Addgene #50005) with primers OF/R49. After gel purification using the Zymoclean Gel DNA Recovery Kit, the purified product was combined with pGGC000 (Addgene #48858) in a Gibson assembly reaction using NEBuilder master mix (NEB). For the final Golden Gate reaction of the LoxPsym combinatorial library, all entries were pooled and entries B and D contained a mix of all LoxPsym variant plasmids at equal concentrations (16 plasmids/position; quantified with Qubit™ dsDNA HS assay) to make a combinatorial library of plasmids containing the 256 different LoxPsym combinations, which was next transformed to DH10B cells. After overnight incubation, the colonies of nine different plates (>50,000 colonies) were scraped and suspended in LB medium. The plasmid DNA was extracted with the ZymoPURE II Plasmid Midiprep Kit (Zymo Research). Plasmids were diluted to 1 µg/µL. The plasmid expressing Cre recombinase was also constructed using Golden Gate with the cloning scheme in Supplementary Table 3, starting from available parts (https://gatewayvectors.vib.be/) and was purified and diluted similarly.

### Z. mays protoplast isolation and transfection

The protocols for the isolation and transfection of maize protoplasts were adapted from Gaillochet et al. (2023)[100]. Maize seeds (cultivar B104) were sown on hydrated Jiffy pellets (No. 32170138, Jiffy Products International). For the first five days after sowing, the trays were placed in a growth chamber at 25 °C, 55% relative humidity under light (16 h light/8 h dark) provided by high-pressure sodium vapor (RNP-T/LR/400W/S/230/E40, Radium) and metal halide lamps with quartz burners (HRI-BT/400W/D230/E40, Radium). Subsequently, the trays with seedlings were transferred to the dark (25 °C, 55% relative humidity) and grown for eight more days. Etiolated maize leaves were harvested, and the middle part of the second leaf was cut into 0.5 mm strips. Leaf trips were then infiltrated with 25 mL cell wall enzyme solution (0.6 M D-mannitol, 10 mM MES, 1.5% cellulose, 0.3% Macerozyme R10, 0.1% BSA and 1 mM CaCl2) using vacuum (50 mmMg) for 30 min in the dark. Then the strips were incubated for 2 h at 25 °C on a shaking platform (0.02688 x *g*) in the dark. The solution was filtered using a sterile

40 µm cell strainer (Corning), and the protoplasts were collected by centrifuging at 100 x *g* (slow acceleration and brake) for 3 min. The supernatant was removed, and protoplasts were washed with ice-cold 0.6 M D-mannitol. The protoplasts were again centrifuged at 100 x *g* (slow acceleration and brake) for 2 min. Protoplasts were resuspended in 5 mL of 0.6 M D-mannitol and incubated in the dark for 30 min. The supernatant was carefully removed, and protoplasts were resuspended in MMG solution (0.6 M mannitol, 15 mM MgCl2, 4 mM MES). After counting the protoplast concentration using a Neubauer chamber, it was adjusted to a concentration of $1 \times 10^6$ cells mL$^{-1}$. Transfections were done in 1 mL strip tubes (TN0946-08B, National Scientific Supply Co), using 100 µL of protoplasts (105 cells), 110 µl of PEG solution (0.2 M mannitol, 100 mM CaCl2 and 40% PEG 4000 (81240, Sigma)) together with 20 µg of plasmid DNA (10 µg of UBI-Cre-NOST or 10 µg of the p35S-mCherry-NLS-AarI-NOST (vector ID 18_15) for the control, and 10 µg of the loxP deletion library). After adding the reagents, protoplasts were incubated for 10–15 min in the dark and 750 µl W5 solution was added to stop the transfection. The protoplasts were then centrifuged at 100 g (slow acceleration and brake) for 2 min, and after discarding the supernatant, the cells were resuspended in 1 ml of W5 solution. Each transfection was done in triplicate. The cells were then transferred to 24-well plates (VWR) and incubated in the dark at 25 °C on a shaking platform (0.00672 x *g*). Samples were harvested after two days and stored at −20 °C until further processing.

### *Z. mays* DNA extraction

A modified Edwards extraction protocol was used for the isolation of protoplast DNA[106]. The extraction buffer was composed of 100 mM Tris HCl (pH 8), 500 mM NaCl, 50 mM EDTA and 0.7% SDS. Protoplasts were transferred to 1.5 mL Eppendorf tube and were spun down at 12,000 rcf for 5 min, after which the supernatant was removed. A volume of 200 µL extraction buffer was added to the Eppendorf tubes and the tubes were manually shaken to dissolve the pellet. After 15 min of incubation at 60 °C, the tubes were cooled down to room temperature. A volume of 200 µL 100% isopropanol was added and the tubes were spun down at 12,000 x *g* for 10 min. The supernatant was removed, and the pellet was washed with 200 µL 80% ethanol. After air drying for 15 min, the pellet was dissolved in 20 µL of 10 mM Tris-HCl pH 8 (preheated at 60 °C). After incubation of the tubes in a 60 °C thermoblock for 10 min, the tubes were stored at −20 °C until further processing.

### Next generation sequencing

For sequencing of the input plasmid library for *Z. mays* transfection, we set up a 40 µL PCR reaction with the Phire Plant Direct PCR Kit (Thermo Scientific) using 4 µL of the diluted midiprep (100 ng/µL) as the template and primers OF/R82. The following PCR conditions were used: 98 °C/2 min + 10 x (98 °C/5 s + 62 °C/5 s + 72 °C/10 s) + 72 °C/2 min + 23 °C/∞. The fragment of the correct size (approx. 270 bp) was purified using the Zymoclean Gel DNA Recovery Kit according to the manufacturer's instructions. A similar set-up was used for the sequencing of the protoplast assay fragments, using 4 µL of the protoplast DNA as the template and primers with different demultiplexing tags for each sample in a total reaction volume of 40 µL. The PCR conditions used were as follows: 98 °C/2 min + 25 x (98 °C/5 s + 62 °C/5 s + 72 °C/10 s) + 72 °C/2 min + 23 °C/∞. We could not detect any evidence of recombination in our agarose electrophoresis results and reasoned that this could be due to the massive amount of plasmid DNA that was transfected (-32 million plasmid copies per protoplast, Supplementary Fig. 9b, c). Therefore, we used restriction-digestion of the extracted DNA to specifically cut the C-linker-D module of the non-recombined plasmids to bias against amplification of these DNA species. Digestion of the protoplast DNA was done with NcoI-HF (NEB) and PvuI-HF (NEB) in CutSmart buffer for 12 h at 37 °C. Amplicons were constructed using primers OF/R83-88 and purified with the GeneJET PCR Purification Kit (Thermo Fisher) according to the manufacturer's instructions. Samples were sent to Eurofins (Germany) for adapter ligation and NGS sequencing (5 million paired reads, 2 x 150 bp). The FASTQ files were imported into Geneious Prime version 2022.2.1 as paired reads and merged using the *BBMerge Paired Read Merger* (Version 38.84) module using standard settings. The merged reads were then demultiplexed by using the Separate Reads by Barcode function in Geneious Prime version 2022.2.1. The demultiplexed files were then downloaded and stored as a .txt file. The number of reads containing the different possible loxPsym-barcode combinations were then counted using a custom Python (version 3.9) script (see Supplementary Fig. 10). Further data analysis was done using Microsoft Excel. For each plasmid, the number of reads detected for the protoplast DNA was normalized by the number of reads present in the input library.

### Statistical analysis

To determine normality of the data, we applied Shapiro Wilk's method. Analysis of statistical differences between the means of samples was done using the Wilcoxon rank sum test, the Kruskal-Wallis test. Alternatively, a linear model was constructed and analysis of variance was performed with post hoc test Tukey honest significant differences test or Dunnettx's multiple comparison of means. To compare the variance of samples, we used a Fligner Killeen test. To fit a linear regression we applied function stat_poly_eq. All statistical analysis were performed using R version 4.3.0 with R packages ggplot2[107], stats, qqpmisc[108] and emmeans[103].

### Reporting summary

Further information on research design is available in the Nature Portfolio Reporting Summary linked to this article.

## Data availability

All data supporting the findings of this study are available within the paper and its supplementary information files. Source data are provided with this paper. NGS and Sanger sequencing data have been deposited to NCBI Sequence Read Archive database under accession code PRJNA1002847. Source data are provided with this paper.

## Code availability

Code for analyses in this study (Phyton version 3.9) is provided in the Supplementary Information File.

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

## Acknowledgements

We thank Dr. Karin Voordeckers for relevant feedback on the manuscript and all Verstrepen lab members for valuable discussions. C.C. was supported by a PhD fellowship from FWO (1S25923N, 1SC2422N). A.Z. acknowledges a PhD fellowship from Vlaams Instituut voor Biotechnologie (VIB). J.Steensels acknowledges financial support from FWO by a postdoctoral fellowship (12W3918N, 12W3921N). J.Smets was supported by VLAIO (HBC.2020.2623) and Y.Z. was supported by a CSC fellowship (202008440364). L.C. and A.G. were supported by a KU Leuven C1 grant (C16/17/006). J.D.S was supported by Ghent University ('Bijzonder Onderzoeksfonds Methusalem project', BOF15/MET_V/004). Research in the lab of K.J.V. was supported by KU Leuven C1 Financing, VIB, VLAIO, FWO and iBOF.

## Author contributions

C.C., J.Steensels, A.G. and K.J.V. conceptualized the study. C.C., T.B.J. and K.J.V. designed the experiments, which were performed by C.C., J.Smets, Y.Z. and J.D.S. Modeling was done by L.C. and C.C. A.Z. contributed to the data analysis. C.C. and K.J.V. wrote the manuscript and all co-authors reviewed the manuscript.

## Competing interests

C.C. and K.J.V. have filed a patent application relating to this work. All other authors declare no competing interests.
