## [Peer Review File · Nature Communications]

Reviewers' Comments:

Reviewer #1:

Remarks to the Author:

This paper describes a set of 16 orthogonal LoxPsym variants that can be used simultaneously in Cre recombination with little cross-reaction. To develop the system, they characterized 63 new LoxPsym sites obtained by editing the spacer sequence and validated 1192 interactions between these variants in the yeast model *S. cerevisiae*. The functions of the orthogonal sites were demonstrated in other species, such as bacterial and plant models, validating the universality of the toolbox. Additionally, the study identified crucial design principles governing LoxPsym sites, essential for mitigating recombinase cross-reactivity.

To enable simultaneous, large-scale gene recombination, previous efforts have been made to expand the orthogonal recombinase toolbox and orthogonal recombinase recognition sites. Nonetheless, the repertoire of orthogonal recombinases and recognition sites remains somewhat limited, and this paper is innovative in that they have performed a relatively systematic screening of Cre LoxPsym recognition sites by altering the first and last three spacer nucleotides and generating a set of 1192 interactions between the different variants. Moreover, the authors extracted design rules for recognition site spacer nucleotides, potentially applicable to the design of other recombinase recognition sites. The major weakness of the paper, and for many papers developing innovative and complex recombinase systems, is missing a practical application. Although the manuscript posits that these orthogonal sites have the potential for simultaneous large-scale gene recombination, it lacks concrete demonstrations. While the designs hold promises for fundamental scientific research, and the derived design principles offer future applications, the paper could significantly enhance its impact by showcasing real-world applications or providing a basic demonstration of simultaneous multiple gene recombination. However, besides the application, the paper probably has done the best for the system design and characterization. I do believe that the paper can make an impact on synthetic biological system design for myriad applications. Therefore, I support publication in Nature Communications after addressing the comments I have listed below.

Major Comments:

Low Recombination Efficiency Rate for Multiplex Genome Engineering: Their experiment demonstrated that the 16 LoxPsym sites can recombine with high orthogonality (Figure 3). However, for most of the sites, the recombinase efficiency is lower than expected, and for the LoxPsym-CAC site, the efficiency is very low. It is uncertain that how low the LoxPsym-CAC efficiency is compared to the control and if there is a statistically significant difference between the LoxPsym-CAC site and the control. In addition, the standard deviations for some other sites are large in this figure (For example, LoxPsym-GAG and LoxPsym-GCG). What are the reasons behind it? And are the results statistically significant? Showing statistics test results between the modified LoxPsym sites and the control might be helpful to further explain this figure.

Cross-Reactivity of LoxPsym sites: In Figure 3d, the authors noted that 9 out of 208 randomly selected red colonies exhibited cross-reactivity. However, it is unclear which pairs of LoxPsym sites are responsible for this cross-reactivity. Further investigation into specific patterns among these 9 cross-reactivity cases would be insightful. Does it indicate that one or two of the sites are more likely to cause cross-reactivity and should be removed from the set? In addition, the author didn't define the effect of the 9 cross-reactivity cases out of 208 cases. Is this good enough for a real genome engineering experiment? Can we call it a success?

The universality of the LoxPsym sites: The paper demonstrates that the 16 LoxPsym sites are also compatible with bacterial and plant cells. However, it is notable that the recombination efficiency in other cell types does not correlate with that observed in yeast cells. Additionally, there is a significantly higher cross-reactivity in bacterial cells, suggesting that the rules identified in yeast cells have limited applicability across different cell types. The authors rationalize this observation by suggesting that orthogonal recombination is not always transferable between prokaryotic and eukaryotic cells. It might be worthwhile to further investigate the universality of these rules within eukaryotes, particularly in mammalian cells (e.g., HEK cells), which are of interest for metabolic pathway studies. Do these recombinase sites function with high orthogonality and efficiency in mammalian cells as well?

Minor Comments:

Resolution of Figure 1(a) is too low and the number inside the figure is vague.

Figure 1(f) and (g): the max of the y axis scale can be adjusted to smaller such that the difference between each condition can be clearer. Especially in 1(f), for A, C, G, T, the numbers are all closed to 1.

Supplementary Figure 6(d): it is hard to tell the recombination activity is completely shut down for CGTAT while the others are not – the AGTAT also looks very low. Probably also include the exact number for each condition would be clearer.

Reviewer #2:

Remarks to the Author:

“Novel orthogonal LoxPsym sites allow multiplexed site-specific recombination in prokaryotic and eukaryotic hosts” is part 1 to a companion paper titled “Combinatorial optimization of gene expression through recombinase-mediated promoter and terminator shuffling in yeast” also submitted concurrently to Nature Communications by the same first and corresponding authors.

In part 1, Cautereels et al. developed a set of 16 orthogonal LoxPsym variants and demonstrated their use for multiplexed genome engineering in both prokaryotes (*E. coli*) and eukaryotes (*S. cerevisiae* and *Z. mays*). Simply, LoxP is an attachment site cognate to Cre recombinase. The LoxP site is a 34 bp sequence comprising two 13 bp inverted repeats that flank a directional 8 bp spacer. The Cre/LoxP recombinase set can facilitate standard deletion and inversion based on canonical attachment site alignments (i.e., aligned and anti-aligned respectively) which has been shown to be true for dozens of recombinases to date. The authors note that a key limitation of the Cre/LoxP recombinase set “is that it relies on a single recombination recognition site. This implies that upon induction of recombination, all recombination sites present in a genome interact with each other in an unpredictable, stochastic fashion”. Again, the concurrent processing of all attachment sites by a given recombinase is known, well-studied, and has been demonstrated to be a “limitation” for dozens of recombinases. The authors argue that a solution to this limitation is to modify the central conserved region of the LoxP attachment site to create putatively orthogonal attachment sites that can be recombined by a single Cre recombinase, again also known; perhaps the most popular demonstration of this was conducted by Roquet et al. nearly a decade ago (*Science*. 2016 Jul 22;353(6297)) though for different recombinases. Moreover, the use an application of recombinases in both prokaryotes and eukaryotes has been thoroughly demonstrated (including the Cre/LoxP set) notable examples include *Nat Biotechnol* 2017;35(5):453-462. While this study (Cautereels et al.) was executed very well, and the writing, data and figures are high quality the paper is not novel in anyway. Moreover, the authors do not make a compelling case to justify the publication of a set of companion papers, when it is clear one paper can tell the whole story much better.

In summary, Cautereels et al. titled “Novel orthogonal LoxPsym sites allow multiplexed site-specific recombination in prokaryotic and eukaryotic hosts” is not suitable for publication in Nature Communications. Specifically, this paper uses a lot of space and discussion to present a study that is straightforward, lacks novelty, and is technically not that difficult to accomplish. In addition, from a fundamental vantage point this paper does not offer any new insight to our understanding of recombinase function. Accordingly, I do not recommend this paper for publication.

REVIEWER COMMENTS to manuscript NCOMMS-23-37020 (Novel orthogonal LoxPsym sites allow multiplexed site-specific recombination in prokaryotic and eukaryotic hosts)

Reviewer #1 (Remarks to the Author):

This paper describes a set of 16 orthogonal LoxPsym variants that can be used simultaneously in Cre recombination with little cross-reaction. To develop the system, they characterized 63 new LoxPsym sites obtained by editing the spacer sequence and validated 1192 interactions between these variants in the yeast model *S. cerevisiae*. The functions of the orthogonal sites were demonstrated in other species, such as bacterial and plant models, validating the universality of the toolbox. Additionally, the study identified crucial design principles governing LoxPsym sites, essential for mitigating recombinase cross-reactivity.

To enable simultaneous, large-scale gene recombination, previous efforts have been made to expand the orthogonal recombinase toolbox and orthogonal recombinase recognition sites. Nonetheless, the repertoire of orthogonal recombinases and recognition sites remains somewhat limited, and this paper is innovative in that they have performed a relatively systematic screening of Cre LoxPsym recognition sites by altering the first and last three spacer nucleotides and generating a set of 1192 interactions between the different variants. Moreover, the authors extracted design rules for recognition site spacer nucleotides, potentially applicable to the design of other recombinase recognition sites. The major weakness of the paper, and for many papers developing innovative and complex recombinase systems, is missing a practical application. Although the manuscript posits that these orthogonal sites have the potential for simultaneous large-scale gene recombination, it lacks concrete demonstrations. While the designs hold promises for fundamental scientific research, and the derived design principles offer future applications, the paper could significantly enhance its impact by showcasing real-world applications or providing a basic demonstration of simultaneous multiple gene recombination. However, besides the application, the paper probably has done the best for the system design and characterization. I do believe that the paper can make an impact on synthetic biological system design for myriad applications. Therefore, I support publication in Nature Communications after addressing the comments I have listed below.

We would like to thank the reviewer for his/her constructive criticism and suggestions for improvement and clarification. In this revised version, we addressed all of the reviewer's comments, and included new experimental data on multiplexed usage of 16 orthogonal LoxPsym sites simultaneously (new version of Figure 3) to further investigate patterns behind cross-reactivity when multiplexing.

Major Comments:

Low Recombination Efficiency Rate for Multiplex Genome Engineering: Their experiment demonstrated that the 16 LoxPsym sites can recombine with high orthogonality (Figure 3). However, for most of the sites, the recombinase efficiency is lower than expected, and for the LoxPsym-CAC site, the efficiency is very low. It is uncertain that how low the LoxPsym-CAC efficiency is compared to the control and if there is a statistically significant difference between the LoxPsym-CAC site and the control. In addition, the standard deviations for some other sites are large in this figure (For example, LoxPsym-GAG and LoxPsym-GCG). What are the reasons behind it? And are the results statistically significant? Showing statistics test results between the modified LoxPsym sites and the control might be helpful to further explain this figure.

We thank the reviewer for this suggestion. We now provided a statistical analysis to Figure 3 (c) by linear regression (using the colony count data as weight for the glm model) and multiple comparison of means using the Dunnett method. With this approach, we compared all samples to both control 1 and control 2. We also added the details of our approach to the methods section (lines 744-751).

With the exception of LoxPsym-CAC, we found that the recombination efficiencies of all variants, also those with high standard deviations, are significantly different from the control groups, with very low p-values (p-values now provided in Supplemental Table 4). Concerning LoxPsym-CAC, the specific reason for the lack of activity in the multiplexed set-up remains unknown. We have now added this information in the text (lines 304-309). Importantly, decreased activity of LoxPsym might also be caused by the general decrease in recombination efficiencies upon multiplexing and the experimental set-up for which we observe that the position within the LoxPsym array plays a significant role for recombination activity (lines 314-323, now included in the main Figure 3, new panel d, where we see that LoxPsym-CAC is quite highly insulated).

The revised text now reads (lines 304-323):

“Interestingly, LoxPsym-CAC showed a very low recombination rate (1.8320 ± 1.410 %) that did not significantly differ from the control constructs (p-values 0.3487 and 0.05595 for comparison to control 1 and 2, respectively), indicating that recombination of this LoxPsym variant was strongly reduced by the presence of the 15 other sites. This severe decrease in activity may not be desirable for multiplexed LoxPsym applications, in which case LoxPsym-CAC could theoretically be substituted by either LoxPsym-TAC or LoxPsym-GAC. The other LoxPsym variants showed higher activities, although the recombination efficiencies were consistently lower and did not correlate well with those calculated from the pairwise interaction assay of Fig. 1 e (Supplemental Fig. 7 a-b). This could at least in part be caused by differences in the experimental setup. In particular, our results suggest that the genomic context of the sites plays a major role because the two sites showing the highest recombination efficiency (LoxPsym-TTA and -TCA) were located at both edges of the LoxPsym array. Moreover, a negative correlation ($R^2 = 0.38$, p-value = 0.01037) can be observed between the recombination efficiency and the distance to the edge of the LoxPsym array, indicating that the efficiency drops because more LoxPsym variants hinder the recombination site of interest to be bound or find its correct interaction partner (Fig. 3 d). We hypothesize that this may be due to a combination of the negative correlation between the recombination efficiency and the distance between interacting recombination sites^{72,73}, the reduced ratio of Cre enzymes and its target site and the formation of nonproductive synapses between incompatible recombination sites, which could shield the recombination sites from recombining with the compatible interaction partner^{62,74}.”

Figure 3 now includes information on p-values after statistical analysis on the data of panel d (see below). Supplemental Table 4 now includes details on this statistical analysis and the obtained p-values.

Cross-Reactivity of LoxPsym sites: In Figure 3d, the authors noted that 9 out of 208 randomly selected red colonies exhibited cross-reactivity. However, it is unclear which pairs of LoxPsym sites are responsible for this cross-reactivity. Further investigation into specific patterns among these 9 cross-reactivity cases would be insightful. Does it indicate that one or two of the sites are more likely to cause cross-reactivity and should be removed from the set? In addition, the author didn't define the effect of the 9 cross-reactivity cases out of 208 cases. Is this good enough for a real genome engineering experiment? Can we call it a success?

We thank the reviewer for pointing out this concern. We fully agree with this comment and decided to repeat the experiment and increase the number of replicates (from 3 to 6) and analyzed clones (from 13 to 36) to obtain a better understanding of the chances of cross-reactivity and see if we can indeed identify patterns within this cross-reactivity (new Figure 3 panels c-d-e-f). Therefore, we sequenced all samples that deviated from the expected fragment size after recombination (54 samples in total, out of 576 samples analyzed for fragment size) and identified all LoxPsym-pairs for which cross-reaction occurred (Fig. 3 e). We observed that two cross-reactions occurred more frequently than the others, and discussed this in the text (lines 335-341). However, when correcting the number of observations for the number of samples analyzed, we find that all cross-reactions were negligible compared to the targeted recombination events. Specifically, we calculated an overall chance of cross-reactivity of merely 0.0839 %, indicating that this set-up is reliable and can successfully be applied for genome engineering. We now added this to the text (lines 335-347 and in the methods section lines 635-645).

The relevant sections now read:

Lines 335-347:

To investigate if we could detect patterns within these undesirable cross-reactions, all 54 fragments were sequenced (**Fig. 3 e, Supplemental Table 5**). Most cross-reactions only occurred a few times, with the exception of the interaction between LoxPsym-CAC & -TTC (24 instances) and LoxPsym-TCT & -TCA (9 instances). Moreover, 21 of the cross-reactions occurred with LoxPsym-TCA, the outermost LoxPsym variant which was used for positive selection of recombination, again suggesting that LoxPsym activity is skewed by the experimental set-up and that the degree of LoxPsym insulation also affects the likelihood of specific cross-reactions to occur. Therefore, we hypothesize that other cross-reactions may have been detected if another layout of the LoxPsym array would have been used. Most importantly, when normalizing the number of observed illegitimate recombination events to the number of potential occurrences, we observe that the level of illegitimate recombination is negligible for each LoxPsym variant (**Fig. 3 f**). Specifically, the overall frequency of illegitimate recombination is 0.0839 %, indicating that this set of LoxPsym variants can be considered fully orthogonal (see Materials and Methods for calculation).

Lines 635-644:

For each test construct, three randomly selected fragments of correct length were sent for Sanger sequencing. All fragments that deviated from the expected length were also sent for Sanger sequencing to determine the occurrence of each illegitimate cross-reaction. Recombination frequencies of all LoxPsym-combinations of the selected red clones were calculated by normalization to the total number of events that could be detected (36 for recombination between identical LoxPsym sites and 576 (the total number of analyzed samples) for cross-reactions between non-identical LoxPsym sites). The overall chance of illegitimate cross-reactivity was determined by the division between the observed number of illegitimate reactions (58) and the total number of cross-reactions that could occur ($120/\text{sample} \times 576 \text{ samples}$).

New figure 3:

Figure 3: Limited cross-reactivity between 16 LoXPsym variants simultaneously. **a.** Each construct (16 test and 2 control) included all 16 LoXPsym variants and an *ADE2* and *URA3* expression cassette. The *URA3* cassette was flanked by LoXPsym-TCA (control to select for recombination). Controls tested 2 locations of the *ADE2* cassette and should not result in *ADE2* deletion. Test constructs differed in the LoXPsym-NNN variant upstream of *ADE2* (pink) and verified cross-reactivity between all sites and recombination efficiency between identical LoXPsym-NNN. LoXPsym variants in the array were separated by 100 bp⁷². Constructs were inserted at the *CAN1* locus of *BY4741* Δ *ADE2* carrying plasmid pSH47-His-Cre or the negative control pSH47-His-Vec. **b.** After 6 h induction, cells were plated on SC+FOA plates (*URA3* deletion with LoXPsym-TCA). Red clones in test strains (deletion of the *ADE2*) were selected for PCR and sequencing. **c.** Percentage of the population with *ADE2* deletion (red phenotype) representing plate counts of six biological replicates, error bars indicate standard deviation. Color indicates pairwise recombination efficiency (Fig. 1 e). Control strains showed a negligible frequency of *ADE2* deletions (0.61 ± 0.70 % and 0.37 ± 0.22 % for control 1 and 2, respectively). Statistics by analysis of variance and two-sided Dunnett's multiple comparisons of means ('***' p-value < 0.001, '.' p-value < 0.1, 'ns' p-value > 0.1, **Supplemental Table 4**) to control 1 (black) and control 2 (grey). No colonies were observed for strains carrying pSH47-His-Vec (**Supplemental Table 4**). **d.** Correlation between recombination efficiency and degree of LoXPsym insulation (distance to the array border). Colors similar to panels a and c. **e.** Measured (dots/triangles) and expected (crosses) length of the recombined construct of 36 randomly selected *URA3* red clones from each strain. Three randomly selected samples of correct size and all fragments of unexpected size (illegitimate recombination) were analyzed by Sanger sequencing (triangles). Color indicates the sequencing result (green if correct, red-scaled if incorrect, in which case the interacting LoXPsym-pair is also indicated). Bars indicate the number of illegitimate recombination events per LoXPsym variant. **f.** Recombination efficiencies of red clones

normalized by the number of possible observations. Source data for this figure are provided as a Source Data file.

The universality of the LoxPsym sites: The paper demonstrates that the 16 LoxPsym sites are also compatible with bacterial and plant cells. However, it is notable that the recombination efficiency in other cell types does not correlate with that observed in yeast cells. Additionally, there is a significantly higher cross-reactivity in bacterial cells, suggesting that the rules identified in yeast cells have limited applicability across different cell types. The authors rationalize this observation by suggesting that orthogonal recombination is not always transferable between prokaryotic and eukaryotic cells. It might be worthwhile to further investigate the universality of these rules within eukaryotes, particularly in mammalian cells (e.g., HEK cells), which are of interest for metabolic pathway studies. Do these recombinase sites function with high orthogonality and efficiency in mammalian cells as well?

We would like to thank the reviewer for his/her suggestion. We understand the reviewer's point that testing the recombination sites in mammalian sites as well would enhance our knowledge further. However, as we have already conducted an extensive study that evaluates our findings in not just one, but three model organisms, we argue that our study is sufficiently comprehensive and that applying our findings in yet another host organism falls out of the scope of this work. In fact, our work already goes much further than many other papers characterizing orthogonal recombination systems, which often only characterize it in just a single host^{1,2,3,4,5,6}. Moreover, we would like to point out that the rules concerning orthogonality are fully maintained across the 2 eukaryotic species that we tested.

Minor Comments:

Resolution of Figure 1(a) is too low and the number inside the figure is vague. Figure 1(f) and (g): the max of the y axis scale can be adjusted to smaller such that the difference between each condition can be clearer. Especially in 1(f), for A, C, G, T, the numbers are all closed to 1.

We thank the reviewer for this comment. We agree with the first part and we simplified Figure 1 (a) and increased font size. For the published version of the paper, all Figures will be provided as .svg files or .png files with enhanced resolution according to the journals' requirements. Resolution of the figure was enhanced from 700 to 1200 DPI for the revised manuscript. We understand the reasoning behind the reviewer's suggestion to enhance the scale of the y-axis for Figure 1(f) and (g). However, we specifically adjusted the scale of all graphs to that of the middle graph of Figure 1(g), which reaches 8 due to the error bar and can therefore not be reduced, to facilitate comparison between all graphs. We prefer to use the same scale for all other graphs as we believe this allows a better comparison between all graphs and facilitates understanding and clarity of the importance of interactions between spacer nucleotides. We do agree with the reviewer's comment to also clarify the differences between the dots for each graph and have now added the graphs with adjusted scales to Supplemental Fig. 3 (panels d and e).

New Fig. 1 panel a:

Figure 1: Effect of LoxPsym spacer sequence on recombination efficiency. a. Cre-LoxP recombination. Cre monomers form a tetrameric complex comprising two active (light) and two inactive (dark) units interacting with two LoXP targets, depicted in black (inverted repeats) and grey (spacer sequence), with the scissile base pairs shown in pink.

New Supplemental Fig. 3 panels d and e:

Supplemental Figure 3: Relationship between LoxPsym spacer sequence and recombination efficiency. d. Magnification of Fig. 1 f showing the effects of single nucleotides on the recombination efficiency for each position of the spacer, calculated by the generalized linear mixed-effects model fit 3 (Supplemental Table 2). Odds ratios > 1 (dotted line) indicate that the event is more likely to occur as the predictor increases, odds ratios < 1 indicate the opposite. Dots and error bars represent the odds ratio and standard error, respectively. Statistics are performed on the log odds ratio scale and represent multiple pairwise-comparison by two-sided Tukey honest significant differences ('***' p-value < 0.001, '**' p-value < 0.01, '*' p-value < 0.05, '.' p-value < 0.1). **e.** Magnification of Fig. 1 g

showing the effects of the interactions between nucleotides at two positions of the LoxPsym spacer. Data and statistics similar to **d**.

Supplementary Figure 6(d): it is hard to tell the recombination activity is completely shut down for CGTAT while the others are not – the AGTAT also looks very low. Probably also include the exact number for each condition would be clearer.

We thank the reviewer for pointing this out. Exact recombination activities of CGTAT and AGTAT are 1.48 ± 0.19 and 3.90 ± 0.37 , respectively. All the exact numbers (for all replicates and all figures) are now provided as a 'source data' file and will be available for readers together with the manuscript and supplemental material upon publication at Nature Communications. We now also refer to the source data file in each figure legend. Therefore, we decided to not add these numbers on top of the graphs because this would reduce the clarity of the figure.

Reviewer #2 (Remarks to the Author):

“Novel orthogonal LoxPsym sites allow multiplexed site-specific recombination in prokaryotic and eukaryotic hosts” is part 1 to a companion paper titled “Combinatorial optimization of gene expression through recombinase-mediated promoter and terminator shuffling in yeast” also submitted concurrently to Nature Communications by the same first and corresponding authors.

In part 1, Cautereels et al. developed a set of 16 orthogonal LoxPsym variants and demonstrated their use for multiplexed genome engineering in both prokaryotes (*E. coli*) and eukaryotes (*S. cerevisiae* and *Z. mays*). Simply, LoxP is an attachment site cognate to Cre recombinase. The LoxP site is a 34 bp sequence comprising two 13 bp inverted repeats that flank a directional 8 bp spacer. The Cre/LoxP recombinase set can facilitate standard deletion and inversion based on canonical attachment site alignments (i.e., aligned and anti-aligned respectively) which has been shown to be true for dozens of recombinases to date.

We understand the reviewer’s point that we may not have sufficiently acknowledged previous research evaluating other recombinases. We have now expanded the introduction of the revised manuscript (lines 52-55). Note that the original version of our manuscript already acknowledged that many site-specific recombinases can be used to facilitate deletion and inversion (lines 31-35). Further along the text, we prefer to keep the major focus of the introduction on the Cre recombinase, as the key message of our paper is the development of new recognition sites for this specific recombinase, together with novel insights on this specific recombination process (see below).

The revised section now reads:

Lines 31-35:

“Site-specific recombination has become a staple tool in today’s molecular biology and genetic engineering in both prokaryotes and eukaryotes (see for example^{1,2,3,4,5,6,7,8,9,10}). Site-specific recombination techniques mostly depend on recombinases that recognize and recombine specific DNA sequences, resulting in deletion, inversion, integration and translocation of large chunks of DNA¹¹.”

Lines 52-55:

“A plethora of different site-specific recombinase systems have been described and generally, all site-specific recombinases fall within one of two groups: serine recombinases (e.g. ϕ C31 and Bxb1) and tyrosine recombinases (e.g. Flp, λ and Cre). Although the recombination mechanism of both groups differs, they both rely on recognition site alignments to enable DNA breakage and repair^{33,34,35.}”

The authors note that a key limitation of the Cre/LoxP recombinase set “is that it relies on a single recombination recognition site. This implies that upon induction of recombination, all recombination sites present in a genome interact with each other in an unpredictable, stochastic fashion”. Again, the concurrent processing of all attachment sites by a given recombinase is known, well-studied, and has been demonstrated to be a “limitation” for dozens of recombinases.

We thank the reviewer for pointing out his/her comment on our ‘problem statement’ in the introduction. We acknowledge that our phrasing may not have been optimal, as site-specificity is indeed not just a “limitation”, but also the key feature enabling specificity. Moreover, we agree that this is the case for dozens of recombinases and that we should not limit our description to the Cre recombinase alone when making this point. Therefore, we adapted our wording in the introduction (specifically at lines 84-89 and 100) to reflect this.

We would like to stress, however, that our systematic screen does reveal novel insights concerning recombination site processing, which have not been shown previously. For instance, we demonstrate a crucial role of the scissile base pairs during processing of the LoxPsym attachment site by the Cre recombinase, as this significantly affects recombination efficiency. Moreover, our results demonstrate that the scissile base pairs strongly impacts illegitimate cross-reactions with other recombination sites, which is an unexpected finding (since this base pair does not participate in strand exchange) and has, to the best of our knowledge, not been shown before for any other recombinase.

The revised text now reads (lines 84-89):

“Due to their high specificity, site-specific recombinases are extremely valuable for genome engineering⁴⁹. However, this site-specificity also implies that upon induction of recombination, all recombination sites present in a genome interact with each other in an unpredictable, stochastic fashion^{33,34,35,49}. This limits the use of site-specific recombinases for genome editing, which often requires multiple, independent, specific genomic edits.”

The authors argue that a solution to this limitation is to modify the central conserved region of the LoxP attachment site to create putatively orthogonal attachment sites that can be recombined by a single Cre recombinase, again also known; perhaps the most popular demonstration of this was conducted by Roquet et al. nearly a decade ago (Science. 2016 Jul 22;353(6297)) though for different recombinases.

We thank the reviewer for raising this point of concern. However, we disagree with this comment for two major reasons.

(1) It has indeed been shown before that altering the Cre recombinase attachment site (and those of other recombinases) enables to generate orthogonal systems and perhaps the best-known examples of these are the lox511⁷, lox5171 and lox2272⁸ sites. However, the systematic analysis performed in our study refutes some of the general assumptions for orthogonality by recombination site modification. For example, we show that only specific edits rather than just any mismatch at the central region of two recombining sites enable orthogonality, and we even identify cross-reactive sites that have more than one discrepancy. Moreover, we show that even base pairs that do not participate in strand exchange affect (and are in fact major determinants of) orthogonality. Therefore, our

comprehensive analysis provides important, relevant new insights into orthogonality and shows that some of the previously established rules were incorrect or, at the very least, incomplete.

(2) The example given by the reviewer³ indeed focusses on different recombinases (each with their own recombination site). We wholeheartedly agree that orthogonal recombinases are useful in many cases^{9,10,11}. However, a major downside of multi-recombinase-usage, with the prospect of multiplexed, genomic engineering in mind, is that it relies on the expression of many heterologous enzymes in parallel. This is not only impractical, but also only feasible for a relatively small number of target sites. Expressing more than a handful of orthogonal recombinases not only imposes experimental problems (e.g. marker occupancy for plasmid maintenance), but is also likely to introduce cellular stress and metabolic cell burden, which could yield undesired mutations or reduced cell fitness^{12,13}. Therefore, expanding the available set of orthogonal recombination sites for a single recombinase is in fact highly valuable for genome engineering and synthetic biology and a set as large as 16 orthogonal recombination sites goes well above and beyond what was available to date, for any recombinase.

Moreover, the use an application of recombinases in both prokaryotes and eukaryotes has been thoroughly demonstrated (including the Cre/LoxP set) notable examples include Nat Biotechnol 2017;35(5):453-462.

We thank the reviewer for this remark. We acknowledge that recombinases are indeed widely used in both pro- and eukaryotes and we already mentioned this at the very beginning of our introduction (lines 31-35). Moreover, we specifically chose to focus on the Cre recombinase in our study, as it is so widely applicable (mentioned on lines 58-59) and therefore might even enhance the value of our presented set of new LoxPsym recombination sites. In fact, we make use of this to characterize our new recombination sites in other organisms as well, adding to the applicability of our toolbox. Additionally, we would like to point out that many high-impact studies characterizing and utilizing orthogonal recombination systems limit their research to just a single host organism^{1,2,3,4,5,6}. We would like to point out that these examples include the mentioned reference Nat Biotechnol 2017;35(5):453-462¹, which is restricted to research in human cells and, again, relies on the simultaneous use of multiple recombinases to obtain orthogonality. Taken together, our work significantly goes beyond previous work on recombinases. We focus on developing orthogonal recombination sites and present a set of 16 orthogonal sites (the largest set reported so far) that is fully orthogonal across two eukaryotes and slightly deviates from orthogonality in one prokaryote. The small discrepancy between both underlies the importance of demonstrating our findings in multiple organisms.

While this study (Cautereels et al.) was executed very well, and the writing, data and figures are high quality the paper is not novel in anyway.

We would like to thank the reviewer for acknowledging the high quality of the writing, data and figures presented in our paper. However, we respectfully disagree that our study lacks novelty and would like to stress that this is supported by the feedback from Reviewer 1, who explicitly mentions that our study is innovative. The novelty of our study lies within the comprehensive, systematic screen of alternative recombination site compatibilities, allowing us to, for the first time ever, present a large set of 16 orthogonal recombination sites that can be recombined by the same recombinase without cross-reactivity, thereby establishing a toolkit that is more practical and offers different possibilities than those relying on the use of different recombinases (see for example the paper that we co-submitted). Moreover, this systematic, high-throughput screen enabled us to obtain the following novel insights:

- We show that the general assumption in the field (and suggested by previous reports) that simply modifying the recombination site spacer sequence is sufficient to impose orthogonal functioning, is NOT correct.
- We reveal an unexpected, yet crucial role of the 1st/8th spacer nucleotide in the orthogonality decision.
- We show that neighboring orthogonal recombination sites strongly affect each other's functioning.
- Using computational modelling, we extracted design rules for recognition site spacer nucleotides, potentially applicable to the design of other recombinase recognition sites (again, fully recognized by reviewer 1).

We now explicitly specified these new insights revealed by our study at the beginning of the discussion (lines 451-456). The revised text now reads (lines 451-456):

“In this study, we present the development and characterization of 63 novel LoxPsym sites in the yeast *Saccharomyces cerevisiae*. We assessed cross-reactivity between a selection of these variants by a pairwise interaction assay and identified a set of 16 orthogonal LoxPsym variants that can be used simultaneously with no or only minimal cross-reaction. Using a systematic approach for LoxPsym modification, our study reveals the following novel insights to Cre-LoxPsym recombination: (i) simply modifying the recombination site spacer sequence is insufficient to impose orthogonal functioning, refuting the general assumption in the field; (ii) the scissile base pairs play an unexpected, yet crucial role in the orthogonality decision; (iii) neighboring recombination sites affect each other's efficiencies and (iv) new design rules to tweak recombination site activity. We demonstrate that the sites can also be used in other species, including *E. coli* and *Z. mays*. Together, these findings dramatically expand the potential of using Cre-LoxPsym as a gene editing technology, especially for cases where recurrent and/or multiplexed recombination is desirable, for example during strain construction in metabolic engineering efforts.”

Moreover, the authors do not make a compelling case to justify the publication of a set of companion papers, when it is clear one paper can tell the whole story much better.

We understand the reviewers comment. However, we strongly disagree that one paper can tell both stories better because both papers convey two independent and distinct messages:

1. Development and characterization of new LoxPsym recombination sites in *S. cerevisiae*, *E. coli* and *Zea mays*, thereby providing a new toolbox for genome engineering in different organisms. Specifically, the new LoxPsym sites allow large-scale, multiplexed integration and genome engineering.
2. Development and characterization of a new tool for combinatorial, multigene expression optimization of metabolic pathways in *S. cerevisiae*, to offer a novel, efficient way to simultaneously generate and evaluate thousands of variants that show different expression levels of a series of different genes (e.g. a metabolic pathway).

Thus, while the first paper reports a new toolbox for genome recombineering that can be used for many different applications across different organisms, the second study presents a solution for an important problem in metabolic engineering and the up-and-coming field of precision fermentation where microbes are used as cell factories for the sustainable production of food, feed, fine chemicals, fuels and pharmaceuticals. Therefore, while the second paper builds upon the first one, we believe that both studies should be seen as independent papers.

We argue that, besides the clearly distinct messages and merging these into one would inevitably result in one of the findings being obscured by the other, fusing the data into a single paper would drastically reduce the quality, impact and visibility of both studies. Our fear is that the new set of LoxPsym sites would remain much more under the radar of the broader research community, while it actually has a huge potential in several fields outside of metabolic engineering and expression tuning and may in fact be the most impactful. That said, we believe it is beneficial to publish both studies back-to-back, because the second study builds upon the LoxPsym sites that are described in the first paper, while in the meantime it also provides a good example of how these LoxPsym sites can be exploited in synthetic biology.

In summary, Cautereels et al. titled “Novel orthogonal LoxPsym sites allow multiplexed site-specific recombination in prokaryotic and eukaryotic hosts” is not suitable for publication in Nature Communications. Specifically, this paper uses a lot of space and discussion to present a study that is straightforward, lacks novelty, and is technically not that difficult to accomplish. In addition, from a fundamental vantage point this paper does not offer any new insight to our understanding of recombinase function. Accordingly, I do not recommend this paper for publication.

Although we highly appreciate the final remarks of the reviewer, we respectfully disagree with this opinion. Firstly, as discussed above, the presented research is in fact novel, offering new insights and proving some of the previous findings to be wrong or incomplete (as also acknowledged by reviewer 1). Concerning the reviewers comment on the technicality, we would like to mention that constructing and analyzing more than a thousand genetically modified strains is in itself technically challenging and that handling and assaying >3.000 strains (replicates) is even more so. In fact, several high-impact reports that perform high-throughput strain construction have been acknowledged for such accomplishments previously^{14,15}.

Finally, we once more want to stress the value of a system relying on only one recombinase and offering up to 16 different orthogonal sites compared to previous approaches that rely on multiple recombinases and offer fewer combinations (as for example in both references mentioned by the reviewer^{1,3}). Specifically, expressing many orthogonal recombinases is not only slow, technically challenging and hugely impractical, it is likely to introduce cellular stress and metabolic cell burden, which could yield undesired mutations or reduced cell fitness. Moreover, our study offers a far greater number of orthogonal recombination sites and thus enables more complex recombination setups (as demonstrated in the companion paper; but also usable in many other setups, for example for multi-step genome engineering or generation of a landing path¹⁶).

References

1. Weinberg, B. H. *et al.* Large-scale design of robust genetic circuits with multiple inputs and outputs for mammalian cells. *Nat Biotechnol* **35**, 453–462 (2017).
2. Durrant, M. G. *et al.* Systematic discovery of recombinases for efficient integration of large DNA sequences into the human genome. *Nat Biotechnol* **41**, 488–499 (2023).
3. Roquet, N., Soleimany, A. P., Ferris, A. C., Aaronson, S. & Lu, T. K. Synthetic recombinase-based state machines in living cells. *Science* **353**, aad8559 (2016).
4. Yarnall, M. T. N. *et al.* Drag-and-drop genome insertion of large sequences without double-strand DNA cleavage using CRISPR-directed integrases. *Nat Biotechnol* **41**, 500–512 (2023).

5. Vo, P. L. H. *et al.* CRISPR RNA-guided integrases for high-efficiency, multiplexed bacterial genome engineering. *Nat Biotechnol* **39**, 480–489 (2021).
6. Guiziou, S., Maranas, C. J., Chu, J. C. & Nemhauser, J. L. An integrase toolbox to record gene-expression during plant development. *Nat Commun* **14**, 1844 (2023).
7. Hoess, R. H., Wierzbicki, A. & Abremski, K. The role of the loxP spacer region in P1 site-specific recombination. *Nucleic Acids Res* **14**, 2287–2300 (1986).
8. Lee, G. & Saito, I. Role of nucleotide sequences of loxP spacer region in Cre-mediated recombination. *Gene* **216**, 55–65 (1998).
9. Short, A. E., Kim, D., Milner, P. T. & Wilson, C. J. Next generation synthetic memory via intercepting recombinase function. *Nat Commun* **14**, 5255 (2023).
10. Liu, W. *et al.* Rapid pathway prototyping and engineering using in vitro and in vivo synthetic genome SCRaMbLE-in methods. *Nat Commun* **9**, (2018).
11. Weng, W., Liu, X., Lui, K. O. & Zhou, B. Harnessing orthogonal recombinases to decipher cell fate with enhanced precision. *Trends Cell Biol* **32**, 324–337 (2022).
12. Borkowski, O. *et al.* Cell-free prediction of protein expression costs for growing cells. *Nat Commun* **9**, 1457 (2018).
13. Frei, T. *et al.* Characterization and mitigation of gene expression burden in mammalian cells. *Nat Commun* **11**, 4641 (2020).
14. Yofe, I. *et al.* One library to make them all: streamlining the creation of yeast libraries via a SWAp-Tag strategy. *Nat Methods* **13**, 371–378 (2016).
15. Meurer, M. *et al.* Genome-wide C-SWAT library for high-throughput yeast genome tagging. *Nat Methods* **15**, 598–600 (2018).
16. Elmore, J. R. *et al.* High-throughput genetic engineering of nonmodel and undomesticated bacteria via iterative site-specific genome integration. *Sci Adv* **9**, (2023).

Reviewers' Comments:

Reviewer #1:

Remarks to the Author:

I found the response to my comments for this and the companion manuscripts satisfactory. They complement each other. Good job.

Reviewer #2:

Remarks to the Author:

The revised manuscript did not address my major concerns - i.e., combining paper one in paper two.

Accordingly, I will leave the final decision with the editor.